# Pre-dauer starvation rapidly and reversibly reduces niche proliferative signaling to the *Caenorhabditis elegans* germ line

Fred A. Koitz, Camille P. Miller, Brian Kinney and Kacy Lynn Gordon*

## ABSTRACT

Early life stresses impact reproductive outcomes in many organisms. In response to crowding and starvation, *Caenorhabditis elegans* nematodes diapause as dauer larvae, in which development arrests until conditions improve. We discovered dramatic differences in gonad size and germ cell number among dauers that form under different conditions. We used live cell imaging of fluorescent proteins in otherwise wild-type and mutant animals combined with food-removal, recovery, and brood-size assays to investigate the causes and consequences of germline differences. Pre-dauer feeding, but not nutrient sensing via the DAF-2/insulin-like signaling receptor or DAF-7/TGFβ, is required for plasticity in gonad size. Gonad differences in dauer have lifelong reproductive consequences; severely starved worms make small dauer gonads and have small broods. Pre-dauer starvation induces germline quiescence and near-instantaneous reduction of the Notch ligand LAG-2 on the germline stem cell niche. A rapid return to germline Notch dependence and an increase in presentation by the germline stem cell niche of LAG-2 – independent of *lag-2* transcriptional upregulation – are among the earliest events of dauer recovery.

KEY WORDS: *C. elegans*, Dauer, Starvation, LAG-2, Notch, Germ line, Germline stem cell niche

## INTRODUCTION

Lifelong development requires environmental responsiveness within the developmental program. A special case of developmental response to difficult environmental conditions is the diapause state, in which development arrests in unfavorable conditions and resumes when conditions improve (Gill et al., 2017; Renfree and Fenelon, 2017). Obligate diapause states are built into the life cycle of an animal in anticipation of poor conditions, while facultative diapause states respond to the environment of an organism (Hand et al., 2016; Wilsterman et al., 2021). One well-studied facultative diapause state is the *Caenorhabditis elegans* dauer (Cassada and Russell, 1975). Dauer larvae can withstand poor environmental conditions (Cassada and Russell, 1975; Golden and Riddle, 1984) for many times the typical lifespan of a worm (Klass and Hirsh, 1976). Improved environmental conditions trigger exit from dauer, resumption of

Department of Biology, The University of North Carolina at Chapel Hill, Chapel Hill, NC 27599, USA.

*Author for correspondence (kacy.gordon@unc.edu)

F.A.K., 0000-0003-2792-7909; B.K., 0000-0001-5628-1436; K.L.G., 0000-0003-0967-4020

development, and a (mostly) normal reproductive life (Cassada and Russell, 1975; Klass and Hirsh, 1976). Famously, time spent in the dauer larval stage does not negatively affect adult lifespan (Hirsh et al., 1976; Klass and Hirsh, 1976).

It is the preservation of fertility rather than increased total lifespan alone that makes dauer adaptive. Previous studies, focusing on recovery, discovered different long-term effects on post-dauer reproduction depending on the dauer-inducing conditions worms experienced (Gimond et al., 2025; Hall et al., 2010; Kim and Paik, 2008; Klass and Hirsh, 1976; Ow et al., 2018; Webster et al., 2018). We set out to investigate the cells of the gonad and germ line during the dauer larval period itself to look for developmental antecedents of later reproductive differences.

*C. elegans* goes through four larval stages (L1-L4) before a terminal molt into the reproductive adult (Corsi et al., 2015). The dauer is an alternative third larval stage. Entry into the dauer developmental pathway is specified in the L1 larval stage by low food and high pheromone, and is followed by L2d, which lasts longer than the standard L2 (Golden and Riddle, 1984). In persistently poor conditions, the L2d molts into a non-feeding dauer larva with a thick cuticle, arrested development including absence of germ cell proliferation, and altered metabolism (Cassada and Russell, 1975; Erkut and Kurzchalia, 2015; Golden and Riddle, 1984; Wadsworth and Riddle, 1989).

The *C. elegans* hermaphrodite gonad develops post-embryonically from a four-cell primordium of two somatic blast cells and two primordial germ cells (Kimble and Hirsh, 1979). In the L1, a distal tip cell (DTC) is born from each of the somatic lineages and the primordial germ cells begin to divide (Kimble and Hirsh, 1979). The germ line relies on DTC expression of LAG-2, a Delta/Serrate/LAG-2 (DSL) ligand of the Notch signaling pathway, to maintain distal germ cells in an undifferentiated state, making the DTC the germline stem cell niche (Kimble and Crittenden, 2005). The developmental arrest of gonad and germ cell lineages in the dauer larva have been investigated genetically (Narbonne and Roy, 2006; Tenen and Greenwald, 2019), primarily using genetic mutants that constitutively form dauers at high temperature, called daf-c mutants.

Among these are mutations of genes in the insulin-IGF-1 signaling (IIS) pathway, including *daf-2*, the gene encoding the sole *C. elegans* IIS receptor (Kimura et al., 1997). IIS integrates nutritional status with metabolism, developmental control, reproduction and aging in *C. elegans* (Ewald et al., 2018; Murphy and Hu, 2013) and in other organisms, including humans (Barbieri et al., 2003; Claeys et al., 2002; Das and Arur, 2017). A 'class 2' reduction-of-function allele, *daf-2(e1370)*, has been used experimentally to induce dauer larvae at high temperature (Gems et al., 1998; Karp, 2018; Kenyon et al., 1993). Another reduction-of-function allele, *daf-7(e1372)* (encoding a TGFβ ligand), affecting a parallel pathway leading to dauer entry, also causes constitutive dauer formation at high temperature (Karp, 2018; Pierce et al., 2001; Ren et al., 1996). The TGFβ pathway links

external cues to the reproductive system (Dalfó et al., 2012; Park et al., 2010, 2021; Pekar et al., 2017).

The germ line is especially sensitive to starvation throughout life (Angelo and Van Gilst, 2009; Dalfó et al., 2012; Pekar et al., 2017; Seidel and Kimble, 2011; Webster et al., 2022), so we hypothesized that pre-dauer nutrition may cause differences in the dauer reproductive system that impact post-dauer recovery of fertility. We discovered that *C. elegans* enter dauer with different numbers of germline cells depending on pre-dauer feeding conditions. More severely starved worms have fewer germ cells and a smaller brood size upon recovery than those that fed more before dauer. Surprisingly, this response was robust to mutation of *daf-2* (IIS receptor) or *daf-7* (TGFβ ligand). The onset of starvation rapidly diminished presentation of the stemness cue LAG-2 by the germline stem cell niche and induced a Notch-independent quiescent state in the dauer germ line. These DTC and germline changes were reversed within hours of recovery from dauer, including exponential recovery of LAG-2 on the DTC that appears to be regulated post-translationally. Altogether, this work reports that differences in recovery of fertility after dauer and dauer gonad variation are mediated by pre-dauer nutrient access, with severely starved worms having small gonads and broods. Pre-dauer feeding affects niche and stem cell maintenance via dynamic regulation of the LAG-2 niche signal at the onset of and upon recovery from starvation.

## RESULTS

### Dauers that form in the presence of food have more germ cells and recover with larger broods than those that form after abject starvation

Different dauer induction regimes lead to differential reproductive recovery (Ow et al., 2018). We compared dauer gonads produced by various regimes. Two of these induce dauer under conditions of abject starvation: dauer entry after food depletion by mixed populations on Parafilm-sealed plates (Karp, 2018) and starvation in liquid culture of a synchronized population (Hibshman et al., 2021). Two other methods produce dauers in the presence of food using crowding (high density plating; Ow and Hall, 2015), or isolation of the first-formed dauers from crowded growth plates (this study; see Materials and Methods). We used an otherwise wild-type marker control strain expressing a germ cell nuclear marker (*mex-5p:: H2B::mCherry::nos-2 3′UTR*) and an endogenously tagged integrin alpha subunit *ina-1(qy23[ina-1::mNG])* that aids visualization of the somatic gonad.

Worms that were starved severely before dauer had significantly fewer germ cells (average of 11 and 15 germ cells for starved plate and liquid culture, respectively) than those that had access to food (average of 22.2 and 37.8 germ cells for high density and first-formed, respectively; Fig. 1A,B), and we recapitulated the finding that dauers induced by high-density plating recover to have larger broods than those that were starved before dauer (Fig. 1C). Brood size of first-formed dauers was also higher than that of starved dauers (Fig. 1C, Table S1A). Different dauer induction protocols changed overall gonad size but did not alter somatic cell numbers in the dauer gonad, only their proximity to one another (Fig. S1A-H), demonstrating that dauer gonads can vary in germ cell number and somatic cell size. The gonad displayed allometric scaling with body width and was relatively smaller in starved dauers compared to first-formed dauers (Fig. S1E), further demonstrating different growth dynamics in the soma and germ line.

We then used the field-standard dauer-formation constitutive (daf-c) mutants carrying reduction-of-function alleles encoding *daf-2(e1370)* (the sole IIS-like signaling receptor) or *daf-7(e1372)*

(a TGFβ ligand), which enter dauer constitutively at high temperature even while feeding (Karp, 2018; Kenyon et al., 1993; Kimura et al., 1997; Ren et al., 1996; Thomas et al., 1993; Vowels and Thomas, 1992). These are not thought to be true 'temperature-sensitive' mutations in which a gene product is functional at low temperature and nonfunctional at higher temperature; rather, it is thought these are reduction-of-function alleles at all temperatures that are sensitized for dauer formation at high temperatures (Dr C. Ewald, ETH Zürich, personal communication; Ewald et al., 2018; Gems et al., 1998; Greer et al., 2008; Kenyon et al., 1993; McGehee, 2019; Trent et al., 1983; Tullet et al., 2008). High temperatures also increase the likelihood of dauer formation in wild-type worms (Ailion and Thomas, 2000). Into these genetic backgrounds we crossed the germ cell nuclear marker shown in Fig. 1A.

At low temperatures, daf-c mutants were able to form dauers using the same protocols that induce dauer in wild-type worms (Fig. 1D-F), and when they did, they displayed the same patterns as wild-type worms: few germ cells when severely starved before dauer, more germ cells when able to feed more before dauer (Fig. 1D-F). When *daf-2(e1370)* and *daf-7(e1372)* mutants were reared at high temperatures and constitutively formed dauer despite feeding *ad libitum*, they had many germ cells (Fig. 1D-F). Thus, daf-c worms, like wild-type worms, have different germ cell numbers in dauer depending on the dauer induction method.

Finally, we measured brood sizes of the *daf-2(e1370)* strain after recovery from dauer induced by different methods. Recovered, starvation-induced *daf-2(e1370)* dauers had lower brood size than recovered crowding-induced dauers, first-formed dauers, and constitutive dauers of that genotype, which all had the same brood size as the never-dauer control *daf-2(e1370)* (Fig. 1G). Our experiments implicate pre-dauer feeding as a potential determinant of germ cell number in dauer and brood size after recovery from dauer, and wild-type alleles of neither *daf-2* nor *daf-7* are required for the response to pre-dauer feeding.

### The germ line grows exponentially with pre-dauer feeding in mutants with defective IIS and TGFβ signaling

To further test the hypothesis that pre-dauer feeding determines dauer germ cell number, we performed a series of food removal experiments on *daf-2(e1370)* and *daf-7(e1372)* daf-c mutants expressing the germ cell nuclear marker. Using daf-c mutants at 25°C ensured that we could vary food exposure while keeping every animal in the L2d dauer entry program. Worms were reared on plates with bacterial food for the specified number of hours at 25°C (Fig. 2A), removed from food at 25°C, and maintained until a population of majority dauers was noted (see Materials and Methods).

We removed worms from food during the L2d stage at 12, 16, 20 and 24 h, and allowed others to feed continuously until dauer (48+ h; Karp, 2018). For each duration of feeding, we assayed germ cell number directly upon removal from food (Fig. 2, Baseline) and again within a day after dauer entry (Fig. 2, Dauer; cuticular alae visible, Fig. S2A,A′). These comparisons allowed us to determine the relationship between time spent feeding and germ cell number at both stages, and how much germ cell numbers changed between the cessation of feeding and dauer entry (Fig. 2A-D, Fig. S2A). Baseline germ cell numbers increased with time spent feeding, and germ cell numbers at terminal gonad size in dauer increased exponentially for both *daf-2(e1370)* and *daf-7(e1372)* (Fig. 2B,C, Fig. S2B,C).

While actively feeding (comparing among 'Baseline' measures in Fig. 2B,C), the gonads of both *daf-2(e1370)* and *daf-7(e1372)* worms grew very little during the interval between 12 and

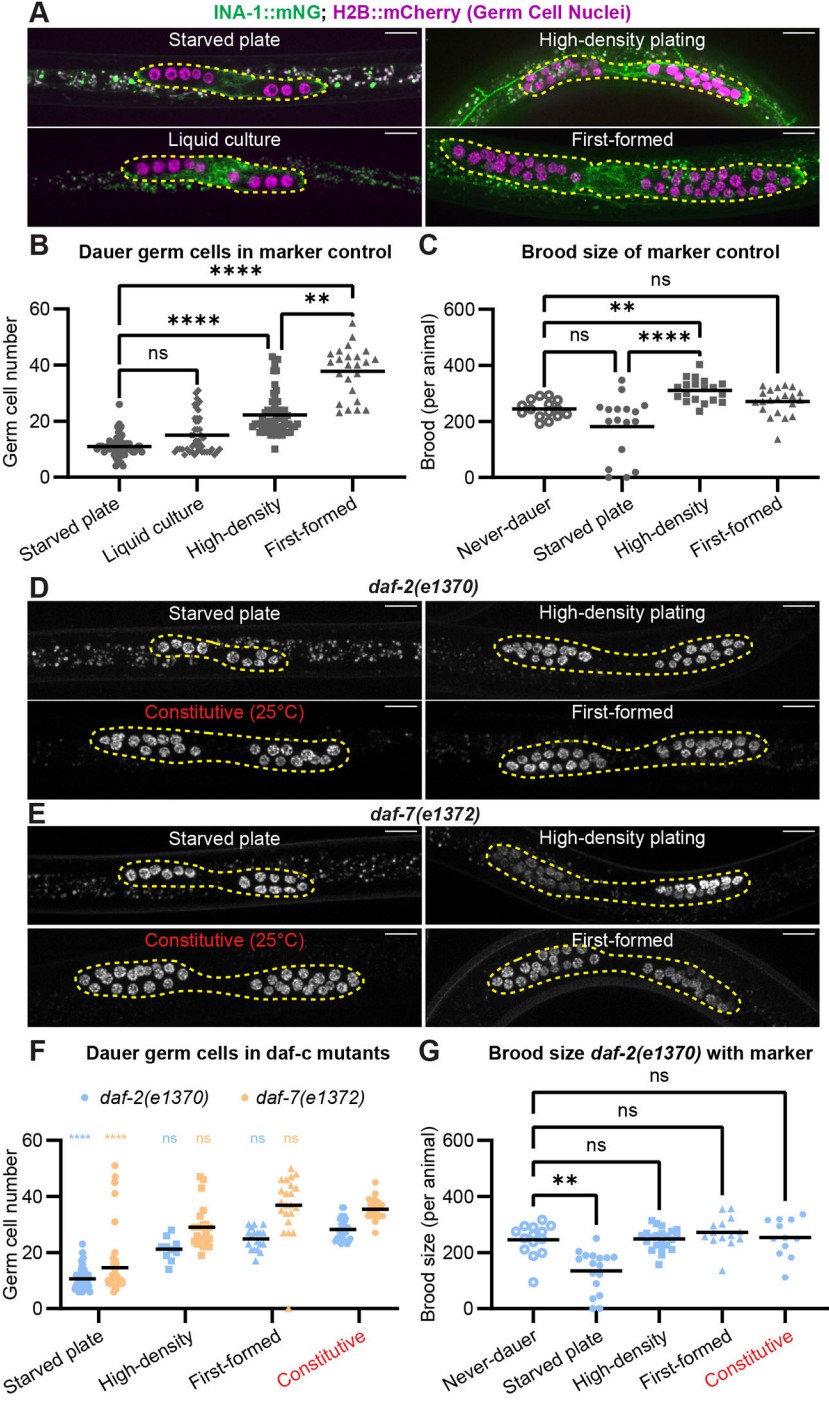

**Fig. 1. Dauer larvae that form after severe starvation have smaller gonads and recover to have smaller broods than dauer larvae that form after feeding.** (A) Representative fluorescence images of dauers induced by indicated methods. Gonads are outlined. (B) Germ cell numbers in dauers of marker control strain induced by different methods: starved plate ($n$=59), liquid culture ($n$=46), high density ($n$=61) and first-formed ($n$=24). Kruskal–Wallis test statistic H=111.0, $P$<0.0001. (C) Brood size of marker control strain. Never-dauer control ($n$=15), recovered dauers from starved plate ($n$=17), high-density plating ($n$=19) and first-formed ($n$=21). Kruskal–Wallis test statistic H=27.19, $P$<0.0001. (D,E) Representative fluorescence images of germ cells in dauers induced by indicated methods for *daf-2(e1370)* (D) and *daf-7(e1372)* (E) expressing a germ cell nuclear marker. Gonads are outlined. (F) Germ cell numbers in *daf-2(e1370)* and *daf-7(e1372)* dauers induced by four methods: starved plate ($n_{daf-2}$=52, $n_{daf-7}$=47), high density ($n_{daf-2}$=37, $n_{daf-7}$=21), first-formed ($n_{daf-2}$=19, $n_{daf-7}$=22) and constitutive ($n_{daf-2}$=33, $n_{daf-7}$=27). Kruskal–Wallis test statistic H=175.3, $P$<0.0001. (G) Brood size of *daf-2(e1370);naSi2(mex-5p::H2B::mCherry::nos-2 3′UTR)* recovered animals under the following conditions: never-dauer control ($n$=14), recovered dauers from starved plate ($n$=17), high-density plating ($n$=15), first-formed ($n$=14) and constitutive ($n$=12). Kruskal–Wallis test statistic H=30.51, $P$<0.0001. Scale bars: 10 µm. Dunn's correction for multiple comparisons was used post-hoc to determine statistical significance of pairwise differences for relevant comparisons. Asterisks on graphs indicate these as ****$P$<0.0001; **$P$<0.005. ns, not significant. Full results of statistical analyses are detailed in Table S1.

20 h (Fig. 2B,C); however, feeding during this stage led to significant differences in terminal gonad size in dauer [for *daf-2(e1370)* mean$_{dauer\ fed\ 12\ h}$=6.68, mean$_{dauer\ fed\ 20\ h}$=10.38, Mann–Whitney U=16.50, $P$<0.0001; for *daf-7(e1372)* mean$_{dauer\ fed\ 12\ h}$=7.16, mean$_{dauer\ fed\ 20\ h}$=17.04, Welch's *t*-test *t*=11.45, $P$<0.0001). Comparing the rate of germline growth for *daf-2(e1370)* animals revealed an even more dramatic response to starvation than that of germ cell number alone (Fig. 2E). Animals gained an average of one germ cell per hour while feeding (12 h fed+24 h fed), while animals that starved (12 h fed+24 h starved) added on average only one germ cell total in that 24 h period (Fig. 2E, gray shaded box) and entered dauer more slowly (Fig. 2F, gray shaded box). Our findings reveal that pre-dauer germline responsiveness to

nutrition is surprisingly robust to mutation in IIS and TGFβ signaling genes.

Average germ cell numbers approximately doubled in both genetic backgrounds between the food removal baseline and dauer entry at each time point (Fig. 2B,C), with the exception of 24 h of L2d feeding for *daf-2(e1370)* animals, which are known to slow germ cell division in late L2d in preparation for dauer entry (Narbonne and Roy, 2006). This slowing may also explain why terminal growth plateaus between 24 h dauers and continuously fed dauers. Germ cell doubling means cells undergo one additional round of division before dauer. Germ cells will complete an average of one additional cell division after DTC laser ablation in the L2-L3 stages of development (Kimble and White, 1981), and genetic

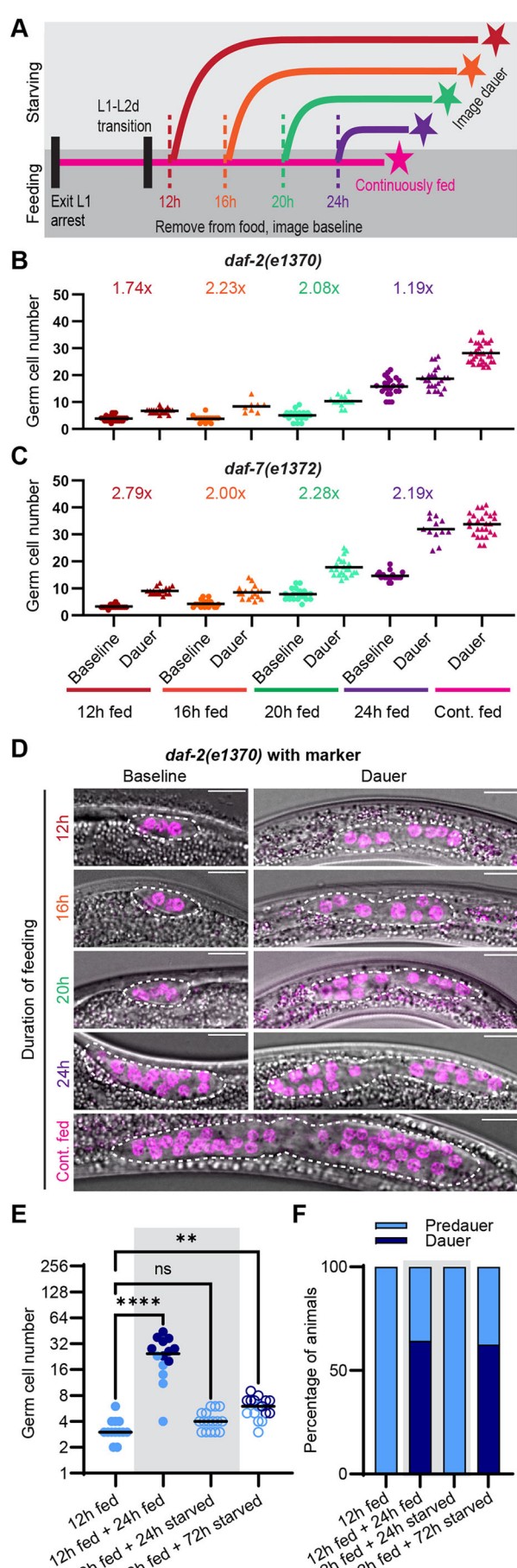

**Fig. 2. Withdrawal of food in the pre-dauer period arrests germ cell proliferation after a single round of division.** (A) Schematic illustrating food removal experiments. Worms were moved off food at different time points (dashed lines) and imaged ('Baseline' in B-D) and then reared until dauer entry (stars) and imaged ('Dauer' in B-D). All experiments were conducted at 25°C. Both strains express a germ cell nuclear marker. (B) Results of the experiment detailed in A, showing the number of germ cell nuclei in *daf-2(e1370)*. Sample sizes: 12 h fed $n_{baseline}$=31, $n_{dauer}$=22; 16 h fed $n_{baseline}$=16, $n_{dauer}$=8; 20 h fed $n_{baseline}$=18, $n_{dauer}$=13; 24 h $n_{baseline}$=23, $n_{dauer}$=24; continuously fed dauer *n*=33. (C) Results of the experiment detailed in A, showing the number of germ cell nuclei in *daf-7(e1372)*. Sample sizes: 12 h fed $n_{baseline}$=20, $n_{dauer}$=17; 16 h fed $n_{baseline}$=20, $n_{dauer}$=17; 20 h fed $n_{baseline}$=20, $n_{dauer}$=22; 24 h fed $n_{baseline}$=18, $n_{dauer}$=12; continuously fed dauer *n*=26. (D) Representative images of *daf-2(e1370)* measured for the experiment shown in B at the time of food removal (baseline; left) and in dauer (right). Gonads are outlined in white. *daf-7(e1372)* shown in Fig. S2. Scale bars: 10 μm. (E) Experiment with *daf-2(e1370)* as shown in A with additional +24 h endpoint. Darker dots indicate animals in dauer diapause at the time of counting. $n_{12\ h\ fed}$=13; $n_{12\ h\ fed,\ 24\ h\ starved}$=16; $n_{12\ h\ fed,\ 24\ h\ fed}$=14 (referring to animals that were washed at 12 h and returned to food for 24 h); $n_{12\ h\ fed,\ 72\ h\ starved}$=16. Kruskal–Wallis test statistic=39.12; *P*<0.0001 with significance from Dunn's correction for multiple comparisons indicated as ****$P$<0.0001; **$P$<0.005. ns, not significant. Note log$_2$ scale. (F) Percentage of worms that had reached the dauer stage under each condition in D. In E,F, the gray shaded box marks the samples quantified in parallel after 24 h under experimental conditions, both on and off food.

ablation of the stemness receptor GLP-1/Notch allows adult germ cells to complete their current mitotic cell cycle to make a terminal division before differentiating (Fox and Schedl, 2015). Based on these established cell cycle constraints on stem-like germ cell proliferation dynamics, our finding that germ cells complete one round of division after food removal suggests that the withdrawal of food effectuates the rapid termination of pro-proliferative signaling to the L2d germ line. The pro-proliferative signal to the germ line is the DTC-expressed LAG-2 ligand, so we next investigated whether this signal was downregulated upon food removal.

### The DTC stemness cue LAG-2 requires *daf-2* and *daf-7* in fed L2s, but not for dramatic downregulation upon food removal in L2d

The pro-proliferative, anti-differentiation signal to the germ line is the DTC-expressed LAG-2 ligand activating GLP-1/Notch signaling in the distal germ cells (Henderson et al., 1994). We wanted to test whether shifting worms off food in the L2d sensitive window in which feeding changes germline growth (shown in Fig. 2) also alters the LAG-2 signal in the DTC. We used an endogenously tagged LAG-2::mNeonGreen and an integrated *lag-2* transcriptional reporter that drives a DTC membrane marker [*qIs154(lag-2p::myr::tdTomato)*] to quantify DTC LAG-2 abundance and *lag-2* transcriptional activity.

Otherwise wild-type dauer worms had notably less expression of both LAG-2::mNG and *lag-2* transcriptional reporter than fed L2 controls, although the magnitude of transcriptional reporter reduction in dauer (25-30%) cannot alone explain the reduction in LAG-2::mNG protein signal (~30 fold) (Fig. 3A-D). Comparing otherwise wild-type worms to *daf-2(e1370)* and *daf-7(e1372)* revealed that both IIS and TGFβ signaling are required for normal *lag-2* promoter activity and normal expression levels of LAG-2::mNG protein in the DTC in fed L2 animals (Fig. 3E-I); this agrees with what is known about the positive transcriptional regulation of the *lag-2* promoter by *daf-7* later in continuous, non-dauer development (Pekar et al., 2017).

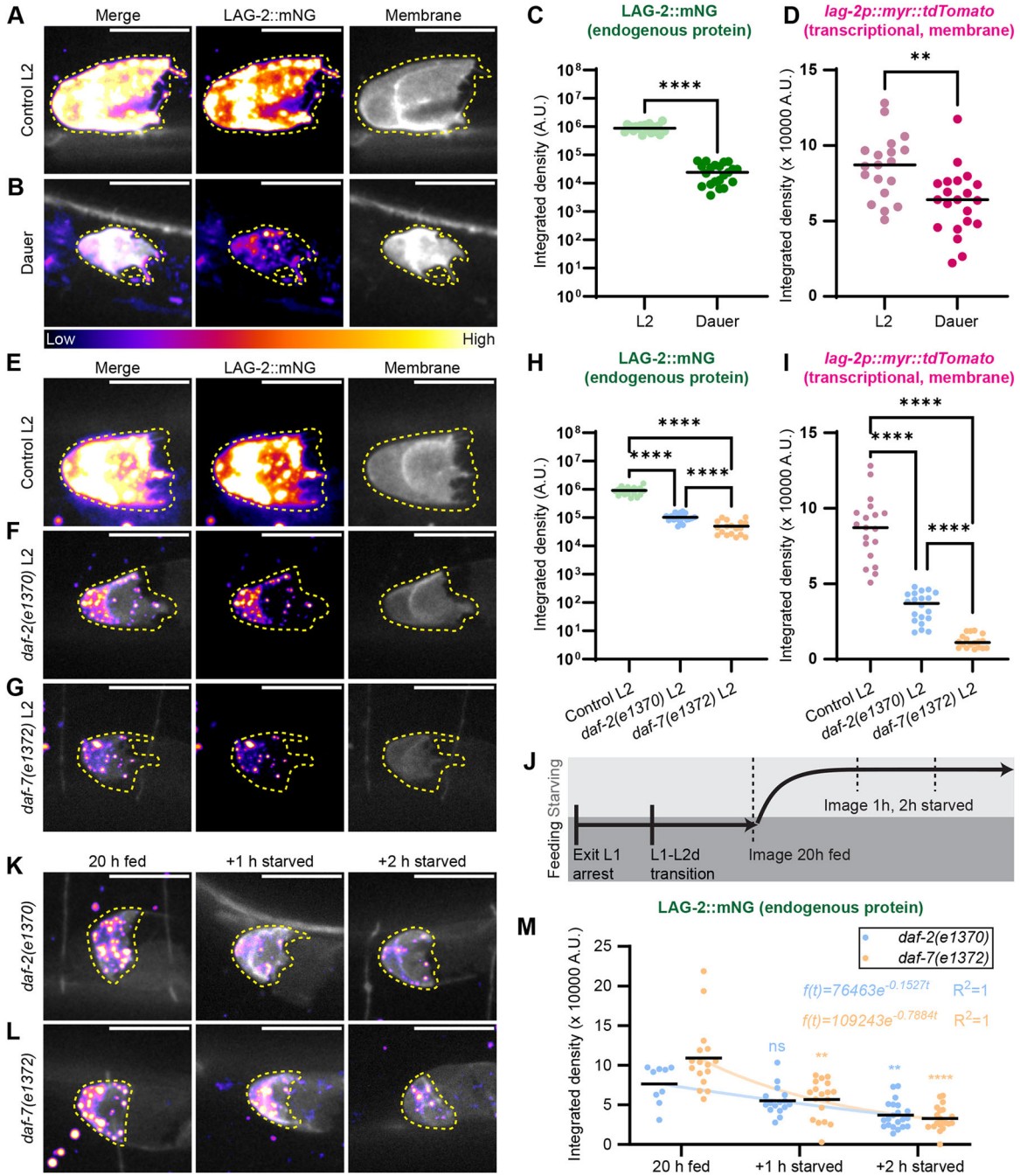

**Fig. 3. LAG-2 expression is higher in fed animals than in dauer and drops rapidly upon food removal in L2d, with *daf-2* and *daf-7* required for the former but not the latter.** (A,B) Sum projections through 11 0.3 µm *z*-slices of the superficial surface of a DTC expressing endogenously tagged LAG-2 protein {fire lookup table; *lag-2(cp193[lag-2:: mNeonGreen])*} and a membrane-localized transgenic reporter [grayscale; *qIs154(lag-2p:: myr::tdTomato)*] in fed L2 controls (*n*=19) and starved plate dauers (*n*=21). Each fluorescence channel is shown with identical scaling between treatments. (C,D) Integrated density measured for the DTCs described in A,B. (C) LAG-2::mNG, note $\log_{10}$ scale. Mann–Whitney test statistic U=0, *P*<0.0001. (D) *qIs154(lag-2p:: myr:: tdTomato)*. Unpaired *t*-test with Welch's correction *t*=3.491, *P*=0.0012. (E-G) Representative images of the transgenes in A in fed L2 worms that are otherwise wild-type (*n*=19) (E), *daf-2(e1370)* mutants (*n*=20) (F) or *daf-7(e1372)* mutants (*n*=18) (G). (H,I) Integrated density measured for DTCs with sample sizes given in E-G. (H) LAG-2::mNG, note $\log_{10}$ scale. Welch's ANOVA test statistic W=89.99 (2.000, 18.76), *P*<0.0001 with Dunnett's T3 multiple comparisons. (I) *qIs154(lag-2p:: myr::tdTomato)*. Otherwise wild-type control, same L2 dataset as in B,C. Welch's ANOVA test statistic W=139.9 (2.000, 29.43), *P*<0.0001 with Dunnett's T3 multiple comparisons. (J) Schematic of the experiment shown in K-M. Worms carrying transgenes in the *daf-2(e1370)* and *daf-7(e1372)* mutant backgrounds were raised on food at 25°C for 20 h, imaged, moved off the food, and imaged 1 and 2 h later. (K,L) Representative images of the worms described in J. Sum projection through 11 0.3 µm *z*-slices of the superficial surface of the DTC at the time points shown above for *daf-2(e1370)* (*n*20 h fed=9, *n*+1 h starved=16, *n*+2 h starved=21) (K) and *daf-7(e1372)* mutants (*n*20 h fed=16, *n*+1 h starved=18, *n*+2 h starved=20) (L) co-expressing endogenously tagged LAG-2 protein (fire lookup table) with DTC membrane-localized transcriptional reporter (grayscale). (M) Integrated density measured for LAG-2::mNG in DTCs with sample sizes given in K,L. Kruskal–Wallis test statistic H=53.78, *P*<0.0001 with Dunn's correction for multiple comparisons; pairwise comparisons were made between 20 h fed and the two food-removal time points per genotype. Regression and $R^2$ values are displayed on the graph. Scale bars: 10 µm. DTCs are outlined in yellow. Significance of pairwise comparisons after post-hoc corrections indicated as *****P*<0.0001; ***P*<0.01. A.U., arbitrary units; ns, not significant.

Next, we compared expression of LAG-2::mNG for each daf-c genotype in a food removal assay in L2d (Fig. 3J). Endogenous LAG-2::mNG protein abundance was highly responsive to the onset of starvation in L2d in both genotypes (Fig. 3K-M). LAG-2::mNG dropped from the L2d 20 h fed baseline within 1 h of food removal in both *daf-2(e1370)* (−27.7%) and *daf-7(e1372)* (−48.1%) mutants, and continued to fall in an exponential decay (Fig. 3M). The abundance of most proteins is governed by first-order kinetics, leading to exponential decay in the absence of new protein synthesis (McShane et al., 2016 and references therein). Thus, our results suggest that LAG-2 deposition on the membrane in the DTC arrests nearly instantaneously upon food removal and existing protein is actively depleted thereafter. The starvation-induced arrest of germ cell proliferation that we observed in the experiments shown in Fig. 2 was accompanied by a rapid starvation-induced downregulation of LAG-2::mNG in the germline stem cell niche, and we conclude that both processes are robust to deficits in *daf-2* and *daf-7* signaling, because we observed them in both reduction-of-function mutants.

This result distinguishes the process observed here from the *daf-7*-dependent downregulation of *lag-2* transcriptional reporters observed in L4 and young adult animals reared in unfavorable conditions (Dalfó et al., 2012; Pekar et al., 2017). Further comparisons of *lag-2* promoter activity and protein abundance in *daf-7(e1372)* mutants reinforced our conclusions that the role of *daf-7* in regulating *lag-2* in pre-dauer and dauer stages (Fig. S3) is distinct from the simple positive relationship reported during continuous development by Pekar et al. (2017). Across stages and conditions, *daf-7* signaling regulates *lag-2* in the DTC in a complex manner, but wild-type *daf-7* is not required for rapid downregulation of LAG-2 in the DTC after the acute onset of starvation (Fig. 3L,M).

### The dauer germ line is maintained in a Notch-independent, quiescent state, and exits that state within hours of dauer recovery

We next examined how the germ line tolerates diminished LAG-2 niche signal in dauer. Under continuous feeding, niche signaling via LAG-2 to the GLP-1/Notch receptor in the germ cells is required for their maintenance in a mitotic, undifferentiated state at all developmental stages (Austin and Kimble, 1989, 1987; Hansen and Schedl, 2006; Henderson et al., 1994; Kimble and Crittenden, 2007; Yochem and Greenwald, 1989). Both DTC ablation (Kimble and White, 1981) and genetic ablation of Notch signaling with a temperature-sensitive allele of *glp-1* (Cinquin et al., 2010; Fox and Schedl, 2015; Kodoyianni et al., 1992) are sufficient to trigger meiosis in all previously mitotic distal germ cells within hours. However, transient adult starvation induces a G2-arrested, Notch-independent, quiescent state in which genetic ablation of Notch signaling does not cause germ cells to differentiate; instead, germ cells remain mitotic upon subsequent recovery of Notch signaling at the permissive temperature (Seidel and Kimble, 2015). Once in the dauer stage, worms are in a *de facto* starved state, and dauer germ cells are G2-arrested (Narbonne and Roy, 2006). We therefore hypothesized that the germ line of dauers, like that of transiently starved adults, is Notch independent for the maintenance of an undifferentiated, stem-like population of germ cells.

We investigated dauer germ cell Notch dependence using worms carrying the temperature-sensitive allele *glp-1(bn18)*. At the permissive temperature (16°C), *glp-1(ts)* mutants have adequate Notch signaling to remain fertile, but upon shifting to the restrictive temperature (25°C), they lose active Notch signaling and thus lose the germline stem cell population to differentiation (Kodoyianni et al., 1992). In adults, signs of meiotic entry appear in distal germ

cells within 6 h of temperature upshift (Fox and Schedl, 2015), so the germ cell fate response to lost Notch signaling is rapid.

We first shifted continuously fed control populations of *glp-1(bn18)* mutants to the restrictive temperature for 24 h (green dashed path in Fig. 4A; see Materials and Methods), then recovered single individuals of various larval stages to plates at the permissive temperature. Nearly all animals recovered as sterile adults (1/28 had a few offspring; Fig. 4B), demonstrating the expected Notch dependence of the fed larval germ line. If the germ line of dauer worms is likewise Notch dependent, sterility should also occur when *glp-1(bn18)* dauers are shifted to the restrictive temperature.

This is not what we found. When we caused *glp-1(bn18)* mutant worms to form dauers at the permissive temperature and then shifted them to the restrictive temperature for 24 h to abolish Notch signaling (purple solid path in Fig. 4A; see Materials and Methods), isolated single dauer individuals to plates with food at the permissive temperature, and allowed the worms to recover, all of the worms recovered as fertile adults (*n*=29/29; Fig. 4B). Thus, the dauer germ line is in a Notch-independent, G2-arrested state and can maintain germline stem cells even in the absence of niche signaling. Because germ cells cannot directly transition from mitotic G2 to prophase I of the meiotic cell cycle without completing mitotic M phase (Fox and Schedl, 2015), cell-cycle quiescence protects dauer germ cells, like those in transiently starved adults (Seidel and Kimble, 2015), from differentiating in the absence of Notch activity.

We hypothesized that the germ line must return to a Notch-dependent state during dauer recovery. We performed temperature-downshift experiments during dauer recovery for *glp-1(bn18)* mutant worms (Fig. 4C; see Materials and Methods). When *glp-1(bn18)* mutant dauers were raised to 25°C, isolated, kept at that restrictive temperature for the first 4 h of recovery from dauer, and then shifted to the permissive temperature to complete development, all animals recovered as fertile adults (*n*=15/15; Fig. 4D), meaning that germ cells remain Notch independent during early dauer recovery. After 6 h of recovery from dauer at 25°C, one-third of worms became sterile in adulthood (*n*=5/15; Fig. 4D). After 8 h of recovery from dauer at 25°C, more than half of worms recovered as sterile adults (*n*=8/15; Fig. 4D). The return of Notch dependence occurs at the earliest in the 4-6 h window after recovery from dauer and is prevalent by 8 h of recovery.

Finally, we found that germ cell proliferation recommences in this same window, with a first doubling observed by 8 h, and modeled at ~7 h by our nonlinear regression (Fig. 4E). This proliferation rate accords with the conservative estimate of doubling in continuously fed L3 germ lines and those recovering from adult reproductive diapause (average of ~9 h; Roy et al., 2016). Mitotic figures reappeared in the germ line during this time period (Fig. 4F,G, indicated by arrowheads in G). The germline mitotic index we observed (2.26% at 4 h and 1.75% at 8 h) is intermediate between the mitotic indices reported for wild-type fed larvae and adults (Roy et al., 2016). The exit of G2-arrested quiescence and resumption of germ cell cycling temporally matches the return of Notch dependence in the germ line. We next examined when during recovery the Notch pathway ligand LAG-2 returns to high levels in the DTC.

### Dauer recovery initiates a transcription-independent burst of LAG-2 ligand presentation by the germline stem cell niche

We predicted that LAG-2 protein abundance would rapidly increase to meet the post-dauer requirement for germline Notch signaling. Immediately after dauer isolation ('0 h' of recovery; Fig. 5A-C), otherwise wild-type dauers had low LAG-2::mNG protein and *lag-2* transcriptional reporter signal. During the first 2 h of dauer recovery,

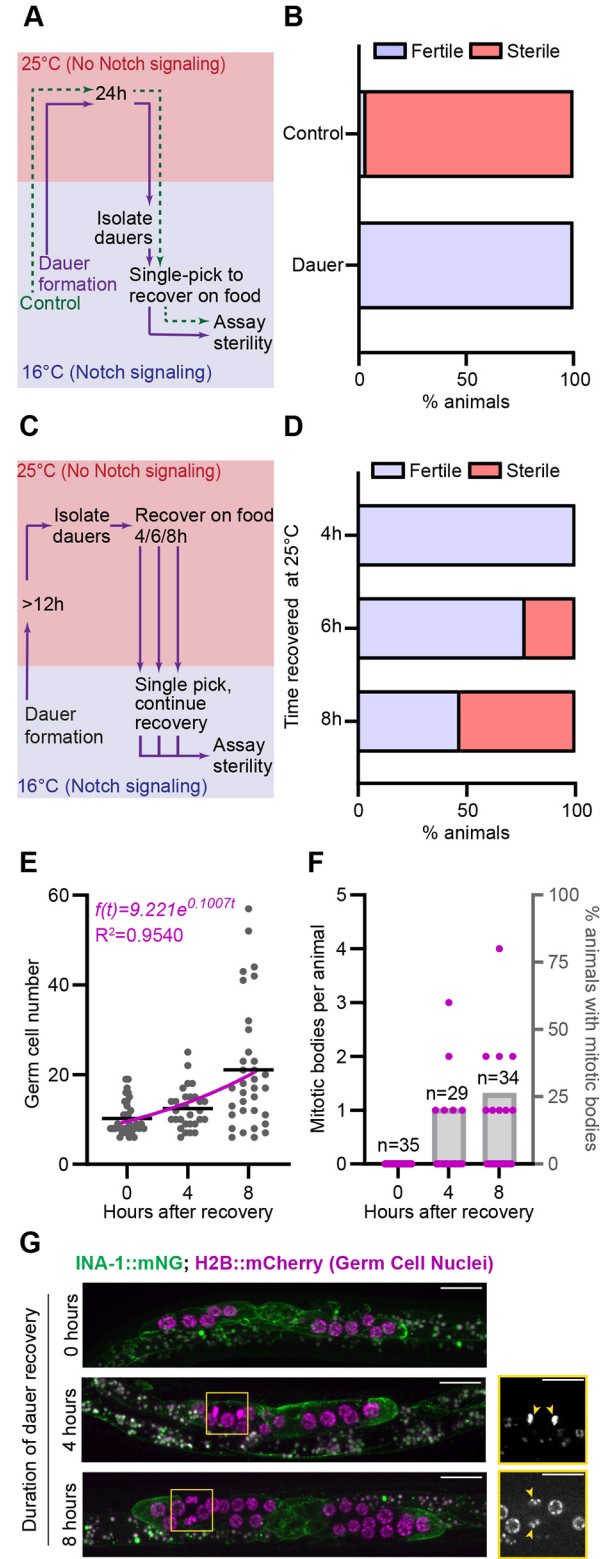

**Fig. 4. Dauer germ cells are arrested in a Notch-independent state and rapidly regain Notch dependence and cell cycling upon dauer exit.** (A) Schematic illustrating temperature-shift experiment testing germline *glp-1* dependence in dauer. (B) Worm fertility after the experiment outlined in A. Control (never-dauer population temperature shift in parallel with dauer) *n*=28; recovered dauers, *n*=29. (C) Schematic of dauer recovery *glp-1(bn18)* temperature-shift experiment testing the return of Notch dependence to the germ line. (D) Percentage of animals recovering as fertile versus sterile after recovery from dauer at 25°C for the first 4 h (*n*=15), 6 h (*n*=15) and 8 h (*n*=15) of dauer recovery. (E) Number of germ cells after the specified hours of dauer recovery in the marker control strain ($n_{0 h}$=35, $n_{4 h}$=29, $n_{8 h}$=34). Regression and $R^2$ value are displayed on the graph. (F) Number of mitotic figures observed in each sample in E. (G) Representative images of gonads at 0 h, 4 h and 8 h of recovery. Note that the 8 h sample has large and small nuclei typical of cells just before and after division, respectively. Boxed areas are shown in single channel on the right. Yellow arrowheads indicate mitotic bodies. Scale bars: 10 µm.

To test this hypothesis further, we examined another pair of *lag-2* reporters. To rule out potential artifacts caused by the multi-copy transgene or slow maturation of myr-TdTomato fluorescence signal, we used a single-copy transcriptional reporter encoding a faster-folding fluorescent protein [*cpIs122(lag-2p::mNeonGreen::PLCδ1^{PH}*)]. To rule out potential artifacts caused by the selection of the 3 kb upstream promoter region for these transgenes and their lack of endogenous transcript regulation, we used a histone H2B::mTurquoise2 knocked into the endogenous *lag-2* locus with a P2A peptide cleavage site {*lag-2(bmd202[lag-2*::P2A::H2B::mTurquoise2^lox511I^2xHA])*}. This element will generate a polycistronic mRNA (Medwig-Kinney et al., 2022), each translation of which will produce one LAG-2 protein and one H2B::mTurquoise2 that separate at translation due to ribosome skipping at the P2A site (Donnelly et al., 2001; reviewed by de Lima and Lanza, 2021). The H2B::mTurquoise2 is therefore subject to endogenous *lag-2* transcript-based regulation (e.g. by transcriptional regulation, 3′UTR-mediated translational repression by microRNAs or RNA-binding proteins, transcript decay, etc.). RNAi against *lag-2* was used to validate this feature of the reagent (Fig. S6).

Up to 8 h after recovery from dauer, we observed no significant change in the levels of expression versus baseline of either the faster-folding single copy transcriptional membrane reporter or polycistronic nuclear reporter (Fig. 5D-F, Fig. S4C,D). Given that the germ line of 53% of animals is Notch dependent by 8 h of dauer recovery (Fig. 4D), transcription-based upregulation of *lag-2* would not appear to be sufficient to increase Notch ligand on the timeline necessary for dauer recovery.

Indeed, the only way to see an increase of endogenously tagged LAG-2::mNG but not an increase in the H2B::mTurquoise2 from the polycistronic knock-in is for the two proteins to be turned over at different rates. We ruled out that H2B::mTurquoise2 is particularly long-lived (see Materials and Methods; Fig. S7). This leaves only LAG-2 protein regulation as a source for their different recovery dynamics. We therefore conclude that the *lag-2* locus is transcriptionally active in the DTC during dauer, despite the germ line being maintained in a Notch-independent state, and that LAG-2 is under protein-specific negative regulation that is rapidly alleviated upon dauer exit.

Taken together, we see an exponential rebound of LAG-2 protein on the DTC begin within 2 h of recovery from dauer, preceding a return of Notch dependence and proliferation in the germ line between 4 and 8 h of recovery. Because dauer exit is asynchronous among individuals (Cassada and Russell, 1975), the precise timing of these events with respect to dauer exit and to one another may be even more tightly coordinated. Other early features of dauer exit – loss of the

LAG-2::mNG protein abundance increased ∼4-fold, and continued to increase exponentially, reaching a 38-fold increase by 8 h post-dauer recovery (Fig. 5A,C, Figs S4A, S5). We did not observe a corresponding increase in signal from the *lag-2p::myr::tdTomato* transcriptional reporter (Fig. 5B,C, Fig. S4B), suggesting that the rapid restoration of LAG-2 protein on the DTC may not be regulated transcriptionally.

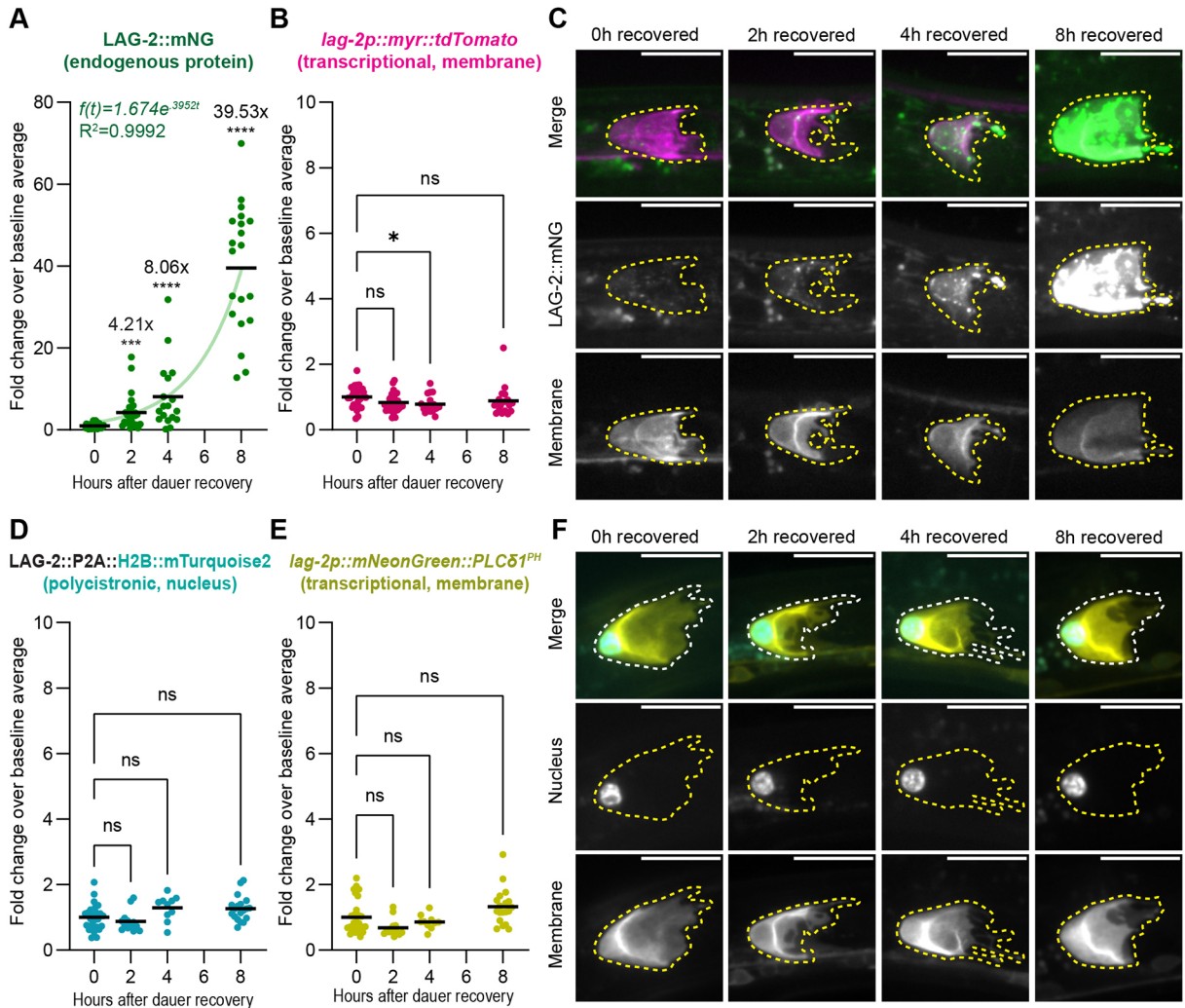

**Fig. 5. Recovery from dauer triggers a burst of LAG-2 presentation on the DTC niche without a coincident rise in *lag-2* transcription.** (A-D) An otherwise wild-type strain co-expressing the endogenously tagged LAG-2 protein *lag-2(cp193[lag-2::mNeonGreen])* (green in C) with a membrane-localized transgenic reporter *qIs154(lag-2p::myr::tdTomato)* (magenta in C) recovering from dauer. (A) Fold change of LAG-2::mNG relative to mean at 0 h ($n_{0\,h}=30$; $n_{2\,h}=29$; $n_{4\,h}=18$; $n_{8\,h}=20$). Note the *y*-axis scale compared to other graphs. Kruskal–Wallis test statistic H=67.26, *P*<0.0001. Dunn's correction for multiple comparisons show all pairwise comparisons are significantly different. (B) Fold change of the *lag-2p::myr::tdTomato* transcriptional reporter for the samples in A. Kruskal–Wallis test statistic H=8.905, *P*=0.0306. (C) Sum projection through 11 0.3 μm *z*-slices of the superficial surface of a DTC for the time points shown above. Each fluorescence channel is shown with identical scaling between time points. (D-F) Otherwise wild-type strain co-expressing a *lag-2(bmd202[lag-2::P2A::H2B::mTurquoise2])* polycistronic histone reporter knocked into the endogenous *lag-2* locus (blue in F) with a *lag-2p::mNeonGreen::PLCδ1^PH* membrane-localized transcriptional reporter (yellow in F). Samples were treated in the same manner as in A-C. (D) Fold change of mTurquoise2 relative to mean at 0 h ($n_{0\,h}=29$; $n_{2\,h}=13$; $n_{4\,h}=10$; $n_{8\,h}=19$). Kruskal–Wallis test statistic H=13.30 m, *P*=0.0040, with Dunn's correction for multiple comparisons showing that none of the recovery time points is significantly different from 0 h baseline. (E) Fold change of *lag-2p::mNeonGreen::PLCδ1^PH* transcriptional reporter relative to mean at 0 h ($n_{0\,h}=29$; $n_{2\,h}=13$; $n_{4\,h}=10$; $n_{8\,h}=19$). Kruskal–Wallis test statistic H=17.41, *P*=0.0006, with Dunn's correction for multiple comparisons showing that none of the recovery time points is significantly different from 0 h baseline. (F) Sum projections through 0.3 μm *z*-slices from the superficial surface of the DTC to the bottom of the nucleus. Non-normalized fluorescence measures and statistical analyses are shown in Fig. S4. DTCs are outlined. Scale bars: 10 μm. ****P*<0.0001; ****P*<0.001; **P*≤0.05. ns, not significant.

buccal plug/SDS resistance and resumption of pharyngeal pumping – also occur within a 2-h window of removal from dauer conditions (Cassada and Russell, 1975), making the return of LAG-2 on the DTC surface as rapid a response to the end of dauer as any other measured event. Our findings establish upregulation of germline stem cell niche signaling as a central, early feature of dauer recovery.

## DISCUSSION

The germ line is the only lineage in *C. elegans* with an indeterminate division pattern and variable cell number, while the somatic lineage is invariant (Kimble and Hirsh, 1979; Sulston and Horvitz, 1977; Sulston et al., 1983). This unique flexibility

allows the germ line to be the only cell population that can respond to starvation by limiting the number of cells it produces, while the soma continues to the 'safe harbor' state of dauer diapause. Starvation induces a precipitate IIS- and TGFβ-independent halt in pro-proliferative niche signaling and germ cell proliferation, which is swiftly reversible upon dauer recovery. That speed is particularly important as germ cells quickly exit the Notch independence of dauer and require active Notch signaling to maintain stemness as they resume cycling during recovery from dauer.

Rapid changes in protein levels and discordance of protein and transcriptional dynamics both suggest post-translational regulation

(Liu et al., 2016). Exponential decay of LAG-2::mNG signal after food removal in L2d in daf-c worms suggests active degradation of LAG-2 without further production. The exponential recovery of LAG-2::mNG upon dauer exit suggests the cessation of that degradation process. We propose a model by which steady but fairly low levels of lag-2 transcription and translation in dauer are coupled with LAG-2 protein degradation triggered by starvation. Upon dauer recovery, protein degradation ceases, leading to the rapid accumulation of LAG-2 protein in the DTC.

Such a regulatory paradigm famously governs the p53 family of tumor suppressor proteins, which are transcribed, translated, ubiquitinated and degraded under normal growth conditions when p53 is not active. DNA damage or other stresses repress ubiquitin ligase, allowing the rapid accumulation, nuclear translocation and activity of p53 (Abuetabh et al., 2022). Recently, cyclin D has been discovered to be regulated by a similar process (Chaikovsky et al., 2021; Maiani et al., 2021; Simoneschi et al., 2021). Other proteins that mediate rapid response to changing environmental conditions are also regulated by steady state production and degradation in the absence of induction, such as the hypoxia inducible transcription factor subunit HIF-α (reviewed by Flick and Kaiser, 2012; Weidemann and Johnson, 2008) and the Nrf2 transcriptional regulator of heme oxygenase HO-1 (Stewart et al., 2003; reviewed by Flick and Kaiser, 2012).

Notch signaling is regulated at the protein level in multiple ways (Fortini, 2009; Suarez Rodriguez et al., 2023; Tian et al., 2004; Wu et al., 2016), including by E3 ubiquitin ligase (Fortini, 2009; Yamamoto et al., 2010). In Drosophila neuroblast stem cells, Notch signaling regulates a nutritionally responsive entry into and exit from quiescence that is driven by amino acid availability, although this mechanism is insulin dependent (Sipe and Siegrist, 2017; Sood et al., 2022). Future work will investigate regulation of LAG-2 in the DTC by the ubiquitin-proteasome pathway, to determine whether it is regulated in a similar way to p53 upon starvation and during dauer exit. Such a regulatory mechanism 'takes the foot off the brake' to rapidly induce a regulator under conditions in which its activity is required, priming a cell to respond to a high-stakes change in state. Protection of the germ line during starvation and recovery is a high-stakes regulatory event in the life of a worm.

Upon food removal, pre-dauer germ cells complete a single terminal division before dauer entry. The most severely starved dauers have as few germ cells (around five to eight) as worms with severe loss-of-function alleles of glp-1 (Austin and Kimble, 1987), and even fewer germ cells than worms have after early laser ablation of the DTC (Kimble and White, 1981), suggesting that the cessation of pro-proliferative cues upon starvation is almost immediate. Thus, we interpret the observation that germ cell number during recovery from dauer differs depending on how dauer was induced (Ow et al., 2021, 2018) to reflect differences in initial dauer germ cell number produced by these same treatments.

The question remains whether a reduction in DTC LAG-2 signal in response to starvation is the proximal cause of germline quiescence, or whether germline quiescence is a necessary precondition for the germ line to tolerate a reduction in DTC LAG-2 as a by-product of starvation. Previous work reported that the dauer germ line is quiescent for germ cell division even with constitutively active Notch signaling (Narbonne and Roy, 2006), suggesting that downregulation of LAG-2 in the DTC is not necessary to prevent germ cell division in dauer itself, although the pre-dauer period we focus on here was not tested. Dauer germ cells are quiescent for proliferation and also for differentiation; meiotic germ cells in glp-1(ts) mutant dauers will not progress through spermatogenesis (Narbonne and Roy, 2006). This

quiescence depends on AMPK signaling (par-4 and aak-2) independent of the DAF-2 IIS receptor (Narbonne and Roy, 2006). The somatic gonad requires DAF-18/PTEN – independent of the DAF-2 IIS receptor or DAF-7/TGFβ – to enter a quiescent state and to mediate germline quiescence in dauer itself, but does not affect quiescence in the pre-dauer period that is the focus of our study (Tenen and Greenwald, 2019).

Expression of both endogenously tagged LAG-2 protein and lag-2 promoter-driven transgenes (Fig. 5; Narbonne and Roy, 2006) remain detectable in dauer, so the downregulation of lag-2 during starvation is relative, not absolute. Downregulation of lag-2 transcriptional reporters in response to starvation or pheromone occurs in later larval stages (Pekar et al., 2017), and the degree of that observed transcriptional difference (~50%) is on the order of what we observe with transcriptional reporters in this study in dauers compared to fed L2s (~30%). In contrast, LAG-2::mNG protein is reduced by ~30-fold in otherwise wild-type dauers compared to fed L2s and then increases by a similar magnitude within hours of recovery from dauer, a window of time in which the signal from transcriptional reporters does not rise at all. Notch signaling via lag-2 is important in dauer and in germline regulation in many contexts (reviewed by Gordon, 2020). Expression of lag-2 in the nervous system is required both during dauer and for dauer recovery (Ouellet et al., 2008), and lag-2 reporter activation in the vulval precursor cells differs between continuous L2 versus L2d and L3 versus dauer (Karp and Greenwald, 2013). LAG-2 protein downregulation in the DTC has been observed during worm aging (Aprison et al., 2024 preprint; Singh et al., 2024), and lag-2 transcription responds to male pheromone signals (Aprison et al., 2024 preprint). All levels of lag-2 regulation – from transcriptional to post-translational – sculpt its dynamics in a range of cells to govern a number of processes, notably those regulating environmental response in the reproductive system.

While the precise molecular mechanisms linking nutrition to LAG-2 protein abundance in the DTC remain to be found, we establish that they function even in the daf-c mutants daf-7(e1372) and daf-2(e1370). The reproductive consequences of pre-dauer food intake are also the same in daf-2(e1370) animals as they are in wild-type worms: pre-dauer starvation leads to reduced brood size. In addition to seeking the molecular mechanisms of starvation-induced LAG-2 depletion in the DTC, future work will also aim to identify the systemic cues that trigger that depletion.

Early life stresses are known to affect later life outcomes in numerous organisms and contexts with potential mechanisms ranging from epigenetics to inflammation to hormone signaling to lasting variation in central nervous system development and function (Taylor, 2010). In C. elegans, primordial germ cells are kept in a quiescent state by the somatic gonad precursors (McIntyre and Nance, 2023). Arrested versus fed L1 worms have differential transcriptional responses in the soma and germ line (Webster et al., 2022), and display chromatin compaction in the germ line (Belew et al., 2021; Morao and Ercan, 2021). The L1 arrest state is triggered by complete starvation after hatching (Baugh and Hu, 2020; Baugh, 2013; Johnson et al., 1984) and several checkpoints triggered by later starvation can arrest somatic development (Baugh and Hu, 2020; Schindler et al., 2014).

Future work will focus on how the germ line transitions from a Notch-independent state back to Notch-fueled germline proliferation. One possible driver is nucleotide levels, which have been shown to limit cell division but not cell growth in unicellular organisms (Diehl et al., 2022) and to affect germline growth in non-dauer C. elegans (Chi et al., 2016). Considering nucleotide levels and the

limits imposed by starvation upon cellular energy prompts the observation that continuous transcription, translation and LAG-2 protein degradation during dauer comes with an energy cost. A dauer worm would only pay this price if the alternative – slower restoration of LAG-2 protein to the germline stem cell niche surface – carried an even greater cost of reduced fertility upon return of the germ line to Notch dependence.

## MATERIALS AND METHODS

Sections of this text are adapted from prior Gordon lab publications (Li et al., 2022; Singh et al., 2024), as they describe our standard laboratory practices and equipment.

### Strains

We used WormBase (Sternberg et al., 2024) and Alliance of Genome Resources (The Alliance of Genome Resources Consortium, 2024) for referencing information about genes, such as genomic location and known patterns of expression. Some strains were provided by the CGC, which is funded by NIH Office of Research Infrastructure Programs (P40 OD010440) and can be requested directly from CGC. The following *C. elegans* strains were obtained from the CGC: N2 (Brenner, 1974), CB1370 *daf-2(e1370)* III (Kenyon et al., 1993; Kimura et al., 1997), CB1372 *daf-7(e1372)* III (Pierce et al., 2001; Ren et al., 1996), DG2389 *glp-1(bn18)* III (Kodoyianni et al., 1992).

The following *C. elegans* strains were generously shared by community members or generated previously in our lab (with sources given for each strain and allele): NK2517 (Gordon et al., 2019) *qIs154(lag-2p::myr::tdTomato)* (Byrd et al., 2014); *lag-2(cp193[lag-2::mNeonGreen^3xFlag])* V; KLG034 (Gordon et al., 2019) *cpIs122(lag-2p::mNeonGreen::PLCδ1^PH)* II (Linden et al., 2017); *lag-2(bmd202[lag-2::P2A::H2B::mTurquoise2^lox511I^2xHA])* V (Medwig-Kinney et al., 2022); GS9692 *arTi435(rps-27p::2xnls::gfp(flexon)::unc-54 3′UTR)* I; *arTi237(ckb-3p::Cre(opti)::tbb-2 3′UTR* X) (Shaffer and Greenwald, 2022).

The following *C. elegans* strains were generated for this paper by crossing existing alleles and markers (with sources given for alleles not previously attributed above): KLG047 *ina-1(qy23[ina-1::mNeonGreen])* (Jayadev et al., 2019); *naSi2(mex-5p::H2B::mCherry::nos-2 3′UTR)* II (Linden et al., 2017; we call this the 'marker control strain' in the text and refer to the *nasi2* transgene as 'germ cell histone marker'); KLG050 *daf-2(e1370)* III; *naSi2(mex-5p::H2B::mCherry::nos-2 3′UTR)* II; KLG051 *daf-2(e1370)* III; *qIs154(lag-2p::myr::tdTomato)* V; *lag-2(cp193[lag-2::mNeonGreen^3xFlag])* V; KLG055 *daf-7(e1372)* III; *nasi2(mex-5p::H2B::mCherry::nos-2 3′UTR)* II; KLG056 *daf-2(e1372)* III*; qIs154(lag-2p::myr::tdTomato)* V; *lag-2(cp193[lag-2::mNeonGreen^3xFlag])* V.

### Strain maintenance and synchronization

Worm strains were maintained on nematode growth media (NGM) at 16°C unless otherwise specified.

### Dauer formation by starvation on Parafilm-sealed plates

To time the formation of facultative dauers more precisely, a large, mixed population of well-fed worms was transferred from an uncrowded growth plate to NGM plates seeded with ~70 µl of an overnight culture of *Escherichia coli* OP50 bacterial food. Plates were sealed with Parafilm and placed at 25°C for the non-temperature-sensitive strains and 16°C for strains containing *daf-2(e1370)*, *daf-7(e1372)* or *glp-1(bn18)* temperature-sensitive alleles. Plates were monitored daily for evidence of starvation (bacterial lawn depleted and worms dispersed). At 25°C, this took ~2-4 days and at 16°C this took ~5-7 days. Plates were analyzed after at least a week of starvation.

### Recovery of first-formed dauers

Dauers were induced as described above until <4 days after food exhaustion was noted, before a robust population of dauers was visible on the plate. Dauers were isolated with a treatment of 1% SDS for 20 min (see 'Dauer isolation protocol' section, below).

### Dauer formation by liquid culture

For dauer formation by liquid culture, the protocol of Hibshman et al. (2021) was used. Briefly, animals were egg prepped (Stiernagle, 2006) from four or five well-populated NGM plates seeded with OP50 and embryos were allowed to hatch out in S-complete medium overnight such that a synchronized population of L1s was generated. Animals were concentrated to five worms per microliter in S-complete and added to glass test-tubes or 25 ml Erlenmeyer flasks, to which *E. coli* HB101 bacteria was added from a concentrated stock such that the final concentration of bacteria in S-complete was 1 mg/ml. Cultures were placed on a shaker (180 rpm). Deviating from the original protocol, animals were incubated at 25°C for 4-5 days and then treated with SDS to match the dauer isolation protocol used in our other methods.

### Dauer formation by high-density plating

For high-density plating conditions, the protocol described by Ow and Hall (2015) and summarized by Karp (2018) was used. Briefly, mixed populations of worms were chunked to four 100 mm plates seeded with 1 ml *E. coli* OP50 or six to eight 60 mm plates seeded with the same titer of food. Plates were grown until densely populated but not depleted of bacterial food. Worms were washed from plates in M9 buffer and allowed to gravity-settle, after which the supernatant (containing younger worms) was removed from the pellet of adults, which was retained for transfer to a single 35-mm plate seeded with *E. coli* OP50. The 35-mm plates were seeded with more bacterial food (100-200 µl of 20× concentrated *E. coli* OP50) than the original protocol (50 µl of an overnight culture) to ensure that the animals had abundant food available prior to dauer formation. The culture plates were then blanketed with a cooked, pasteurized egg white mixture as per the original protocol. To these plates, the pelleted adults were added. For daf-c strains at 16°C, plates were incubated for longer than the 72 h specified by the original protocol, as worm development is slower at that temperature. To recover dauers, plates were SDS-treated when there were dauers visible in a sample of the egg white mixture.

### Dauer isolation protocol

For plates with mixed populations, dauers were isolated with 1% SDS (Karp, 2018). After SDS treatment, the population of recovered animals was placed onto an unseeded NGM plate and the excess liquid was allowed to dry, after which animals were picked to a slide for imaging or picked to an NGM plate seeded with OP50 for recovery and brood measurement. For synchronous daf-c populations in dauer, worms were picked from all-dauer populations and dauer morphology was ascertained for each specimen by microscopy.

### Daf-c mutant constitutive dauer formation

Constitutive dauer formation of daf-c mutants [*daf-2(e1370)* and *daf-7(e1372)*] was achieved by collecting embryos via a standard egg prep (Stiernagle, 2006) with embryos hatching into L1s in M9 to create a synchronized population. This population was concentrated and added to NGM plates seeded with *E. coli* OP50 at 25°C. The mutant daf-c strains used in this study were the reduction-of-function alleles of the insulin-like receptor, class II allele *daf-2(e1370)*, and the TGFβ ligand *daf-7(e1372)*, which are field-standard genetic backgrounds for studying dauer (Karp, 2018).

### Brood size assays

All brood size assays were conducted at 16°C to match the necessary conditions of the daf-c strains. After experimental treatment (see below), worms were singly picked to NGM plates seeded with ~70 µl *E. coli* OP50. Egg-laying adults were subsequently passaged to a new plate once daily until egg laying stopped. Plates with progeny were kept at 16°C until the oldest worm on the plate had reached L4, after which they were counted. Brood totals include broods of animals that died during recovery and animals that were fully sterile, and did not include animals that failed to recover from dauer to adulthood.

For 'never-dauer' control brood assays, fed L4 animals of each strain were singled from mixed populations on day 0. For brood after recovery from constitutive dauer formation, healthy plates of adults were egg prepped; eggs were rolled in M9 overnight at room temperature to obtain a population

of arrested L1 larvae. Animals were dropped onto NGM plates seeded with OP50. Plates were sealed with Parafilm and put at the restrictive temperature (25°C) to induce dauer formation. Worms were then isolated by SDS treatment (see previous section 'Dauer isolation protocol') and dauers were singly recovered to NGM plates seeded with OP50 at 16°C to recover to reproductive adulthood.

Brood size was not measured for *daf-7(e1372)* mutant strains because these worms have a high rate of bagging, even at the permissive temperature (Karp, 2018; Shaw et al., 2007; Trent et al., 1983).

### Food removal experiments

A synchronized population of *daf-2(e1370); naSi2(mex-5p::H2B::mCherry::nos-2 3′UTR II)* L1s or *daf-7(e1372); naSi2(mex-5p::H2B::mCherry::nos-2 3′UTR II)* was obtained via an egg prep of at least eight 60 mm NGM plates seeded with *E. coli* OP50. Synchronized L1s were added to NGM plates seeded with OP50 and were put at 25°C to induce constitutive dauer formation. After the specified number of hours (Fig. 2A), animals were rinsed off the seeded plate with M9, recovered to a 15 ml conical tube, and were subsequently washed in M9 three to five times and then rolled in an excess of M9 at room temperature for 20 min before being washed an additional time in M9 to clear bacteria from the skin and gut. To circumvent worms sticking to the side of the conical tube, tubes were briefly vortexed between spins as needed.

Washed animals were transferred to peptone-free plates (recipe taken from Eustice et al., 2022) and returned to 25°C to continue to develop. Imaging was carried out immediately after animals were washed off the food plates (baseline), and parallel populations were kept at 25°C until the plate visibly had >50% dauers (the rest being pre-dauers; these strains do not progress past dauer at 25°C). Animals that appeared to be dauers under a dissecting scope were picked to be imaged for the dauer endpoint of the experiment; only animals with dauer morphology (recognized by radial constriction and the presence of alae) were measured for germ cell number. While removal of food in the L1 stage prior to the L1-L2d transition caused L1 developmental arrest, as expected, we found that removal of food after the L2d molt did not inhibit *daf-2(e1370)* or *daf-7(e1372)* animals from progressing through L2d or entering dauer.

At this temperature, these strains experience a ~100% developmental arrest in the dauer stage and will not molt beyond dauer unless returned to lower temperatures (Gems et al., 1998; Karp, 2018; Kenyon et al., 1993; Ren et al., 1996). We used this endpoint rather than a set clock-time because the duration of the pre-dauer period of daf-c mutants at 25°C is much longer than the duration of the L1 and L2 stages of fed animals. The *daf-2(e1370)* mutant takes approximately 80 h to enter dauer, and *daf-7(e1372)* takes approximately 48 h (Karp, 2018), compared to just over 24 h for continuously fed animals to reach L3 (Byerly et al., 1976). The pre-dauer period is also extended in wild-type animals that feed on a lower titer of bacterial food (Cassada and Russell, 1975).

### *glp-1(ts)* temperature-shift experiments

Temperature-shift experiments used the temperature-sensitive allele of the Notch receptor *glp-1(bn18)*. Loss of Notch signaling causes failure of germline induction (if the temperature shift happens in the embryonic period) or irreversible loss of the stem-like cell fate and meiotic entry of all germ cells (Fox and Schedl, 2015; Kodoyianni et al., 1992). To test for *glp-1* dependence during dauer (Fig. 4A,B), two populations were used. As a control, an unstarved, uncrowded population was reared at 16°C, shifted to the restrictive temperature of 25°C for 24 h, and larval worms were singly picked to NGM plates seeded with *E. coli* OP50; these plates were kept at 16°C until adulthood. The experimental group was made of worms that had been starved on plates at 16°C to form dauers, which were shifted to 25°C for 24 h. After 24 h, dauers were recovered by SDS isolation and singled to NGM plates seeded with *E. coli* OP50 and kept at 16°C until adulthood. Each plate was subsequently assayed for the presence of progeny.

To test for when *glp-1* dependence returns after dauer (Fig. 4C,D), *glp-1(bn18)* worms were put into dauer on starved plates at 16°C (n=14 first formed and n=15 starved), shifted to 25°C for a minimum of 12 h, SDS isolated to trigger dauer exit, and transferred to NGM plates with food and placed back at 25°C for an additional 4, 6 or 8 h. At that time, the

developmentally oldest/largest worms on the plate (the worms that exited dauer first, since dauer exit is asynchronous) were picked as single animals to NGM plates seeded with *E. coli* OP50 food and kept at 16°C until adulthood. After 4 days of recovery at 16°C, each plate was assayed for live larval offspring, and those that had offspring were scored as fertile. Parental worms on plates that lacked larvae were imaged at 60× magnification and scored for the presence of embryos in the uterus (fertile) or an absence of gametes (sterile). Embryo retention appeared to be a phenotype of *glp-1(bn18)*.

### Confocal imaging

All images were acquired at room temperature on a Leica DMI8 with an xLIGHT V3 confocal spinning disk head (89 North) with a 63× Plan-Apochromat (1.4 NA) objective and an ORCAFusion GenIII sCMOS camera (Hamamatsu Photonics) controlled by microManager. RFPs were excited with a 555 nm laser; GFP and mNG were excited with a 488 nm laser; mTurquoise was excited with a 445 nm laser. Z-stacks through the gonad were acquired with a z-step size of 0.3 or 0.5 μm as noted. Worms were mounted on agar pads in M9 buffer with 0.01-0.02 M sodium azide paralytic [VWR (Avantor) 26628-22-8]. Samples used for fluorescence quantification were acquired with the same laser power, exposure time and z-step size within the datasets to be compared, with histograms monitored to ensure proper exposure. Images with poor body placement, debris on the slide, or other image quality concerns were not analyzed.

### Image analysis software

Images were processed in Fiji (version 2.14.1/1.54f).

### Dauer recovery experiments

To measure fluorescent protein expression and germ cell proliferation during dauer recovery, worms were isolated as described above ('Dauer isolation protocol'), and subsequently split such that some worms were imaged immediately ('0 h of recovery'; Figs 4,5), and other worms were recovered to *E. coli* OP50-seeded NGM plates for the specified length of time.

### Germ cell proliferation during dauer recovery

Our otherwise wild-type marker control strain co-expressing *ina-1(qy23[ina-1::mNeonGreen]); naSi2(mex-5p::H2B::mCherry::nos-2 3′UTR)* II was put into dauer by starvation on plates, and dauers were isolated as above ('Dauer isolation protocol'). Worms were imaged at that 0 h starting point, and parallel populations were recovered to plates seeded with *E. coli* OP50 bacterial food. Worms were imaged at 4 and 8 h after plating on food. Germ cells were counted manually and scored for mitotic figures (metaphase and anaphase chromatin condensations visible in the H2B::mCherry signal; Fig. 4G). We plotted germ cell numbers over time and the percentage of worms for which we observed mitotic figures in Fig. 4F. Mitotic indices were calculated as number of mitotic cells/total cells for each specimen as described by Roy et al. (2016) and averages for each time point are reported in the text.

### Fluorescence intensity measurement of reporters of *lag-2*
#### Endogenously tagged LAG-2::mNG with co-expressed membrane TdTomato

Strain NK2517 *qIs154(lag-2p::myr::tdTomato); lag-2(cp193[lag-2::mNeonGreen^3xFlag]) V* was used to measure endogenously tagged LAG-2::mNG protein (Gordon et al., 2019) and a co-expressed multicopy integrated transcriptional reporter of the *lag-2* ~3 kb upstream promoter with an *unc-54* 3′UTR (Byrd et al., 2014). All worms, including controls, were reared at 25°C. TdTomato has a half-time to maturation of 1 h at 37°C (Shaner et al., 2004), while mNeonGreen has a half-time to maturation of less than 10 min at 37°C (Shaner et al., 2013) (both will fold slower at worm-rearing temperatures). TdTomato signal is visible in adjacent neurons (Figs 3, 5) and the autofluorescent gut granules of the worm are visible in the channel of the endogenously tagged LAG-2::mNG. These structures were avoided as much as possible during projection, tracing and measurement of signal.

We chose an L2 control for this experiment because, while the dauer larva is an alternative third larval stage, the worm reproductive system undergoes morphogenetic changes and changes in cell number during the normal L3 stage. These developmental events are arrested in dauer (Tenen and

Greenwald, 2019), making L3 a confounding control for the dauer larval reproductive system.

Confocal z-stacks were acquired with a 0.3 µm step size. A sum intensity z-projection was created in Fiji consisting of 11 slices representing the superficial half of the cell (from the superficial cell surface to the cross-section of the nucleus, which is visible as a dark void in the center of the cell body). Usually, one DTC from each worm was analyzed, as the gut often obscures one. The DTC was hand-traced on the membrane marker RFP channel; that region of interest (ROI) was subsequently used to measure both the RFP (membrane) and GFP (endogenously tagged protein) channels, and slid off the DTC to the adjacent body of the worm to measure the background for both channels. Background subtracted 'integrated density' measurements were thus obtained. We then normalized all integrated density measurements to the mean integrated density at the 0 h time point for that fluorescent protein, so fold-change values and statistics are shown in Fig. 5A,B,D,E, with raw values shown in Fig. S4. Kruskal–Wallis test with follow-up Dunn's correction for multiple comparisons of the raw, non-normalized integrated densities showed the same pattern of significance (that is, only LAG-2::mNG signal increases relative to baseline) as the fold change analysis (Fig. S4). To display the ~40× range of LAG-2::mNG intensity with the same image scaling, some pixels are saturated in display images (Fig. 5C). Saturation was not observed in the raw images from which measurements were made (Fig. S5).

### Polycistronic histone reporter with co-expressed membrane mNeonGreen

We considered the possibility that the ~3 kb upstream *lag-2* promoter fragment used in the transcriptional reporter transgenes could insufficiently capture endogenous transcriptional regulation, or that post-transcriptional regulation of *lag-2* mRNA could uniquely affect LAG-2 production. The membrane-localized transcriptional reporters that we used from Byrd et al. (2014) and Linden et al. (2017) like those of Blelloch et al. (1999), Henderson et al. (1994) and Pekar et al. (2017) are driven by ~3 kb upstream of the *lag-2* transcription start site (including the TGFβ-responsive element described by Pekar et al., 2017). This ~3 kb element is sufficient to drive expression in DTCs; however, Karp and Greenwald (2013) identified regulatory elements of *lag-2* as far away as ~6.4 kb upstream, raising the possibility that ~3 kb *lag-2* reporters are missing relevant regulatory information that is important for DTC expression under certain, previously unexamined conditions.

We therefore examined a histone H2B::mTurquoise2 (half-time to maturation of 33.5 min; Goedhart et al., 2012) that was knocked into the endogenous *lag-2* locus with a P2A peptide (Medwig-Kinney et al., 2022). This element will generate a polycistronic mRNA, the translation of which will produce one LAG-2 protein and one H2B::mTurquoise2, meaning that the histone reporter is under the same transcriptional and post-transcriptional, transcript-based regulation (e.g. by 3′UTR-mediated repression by microRNAs or RNA-binding proteins, transcript decay, etc.) as the endogenous *lag-2* gene. It is co-expressed with another DTC-expressed transcriptional reporter, a faster-folding, membrane-localized *lag-2p::mNeonGreen::PLCδ1^{PH}* driven by the ~3 kb upstream *lag-2* promoter with a *let-858* 3′ terminator sequence, integrated into the MoscI site on Ch. II (*cpIs122*; Linden et al., 2017). This strain was previously analyzed by Singh et al. (2024) and Li and Gordon (2025).

Confocal z-stacks were acquired with a 0.3 µm step size. A sum intensity z-projection through a 6-7 µm depth of the DTC (to capture the entire nucleus in the z-plane, ~20 slices) was generated in Fiji, the DTC was hand-traced, and membrane fluorescence was measured for the GFP channel in that ROI. A background measurement was made by sliding the DTC ROI onto the adjacent body of the worm. The nuclear histone signal was obtained using an ellipse tool tight around the nucleus in the same z-projection to measure on the CFP channel, and background for the nuclear signal was made by sliding that ROI onto the adjacent body of the worm. Background-subtracted integrated density measurements were thus obtained. We then normalized all integrated density measurements to the mean integrated density at the 0 h time point for that fluorescent protein, so fold-change values and statistics are shown in Fig. 5A,B,D,E, with raw values shown in Fig. S4. Kruskal–Wallis test with follow-up Dunn's correction for multiple comparisons of the raw, non-normalized integrated densities show the same pattern of significance (that is, no pairwise difference relative to baseline for either fluorescent protein) as the fold change analysis (Fig. S4).

To test whether perdurance of H2B::mTurquoise2 could mask the addition of new protein upon transcriptional upregulation (Fig. 5D), we examined its dynamics in the Z1 and Z4 lineages in early larval stages (Fig. S7). Late L1 and early L2 worms (12 h after release from L1 arrest at 25°C), and later L2 (15 h) worms were imaged as above and sum projections were made through ~3.3 µm (11 slices) of the superficial gonad arm (and for one sample, for both arms). This captured the DTC (Z1.aa or Z4.pp) and its sister cell (Z1.ap or Z4.pa), or else caught the parent of these cells (Z1.a or Z4.p) right before it divides. We did not note any systematic difference between Z1- and Z4-lineage cells for expression of this marker despite notably different membrane fluorescence in these cells driven by the ~3 kb upstream *lag-2* promoter (Singh et al., 2025). For each gonad arm, we used an elliptical ROI to measure H2B::mTurquoise2 signal in the nuclei of these cells, as well as a background region in the middle of the gonad where no nucleus expresses the marker. The nucleus of the sister cell was identified for measurement as a void in the GFP channel if no histone signal could easily be noted. Background-subtracted measurements were made and compared to quantify perdurance of the fluorescent histone signal.

### *lag-2* RNAi

To verify that the H2B::mTurquoise2 is indeed co-regulated at the transcript level with endogenous *lag-2*, we measured the effect of *lag-2* RNAi on H2B::mTurquoise2 signal (Fig. S6). Single-colony overnight cultures were made of *E. coli* HT115 containing either the empty vector L4440 or clone Y73C8B.4 from the Ahringer library (Kamath and Ahringer, 2003; Kamath et al., 2003), which is complementary to >900 nt of the *lag-2* mRNA extending from just upstream of the start codon through exon 1, intron 1 and some of exon 2. Cultures contained ampicillin [100 µg/ml, VWR (Avantor), 76204-346] and were grown at 37°C, expression was induced with 1 mM IPTG (Apex BioResearch Products, 20-109) for 1 h at 37°C, and the culture was plated and allowed to grow at least overnight at room temperature on NGM plates prepared with IPTG and ampicillin. L2 larval stage worms were picked onto NGM plates without food and allowed to move around for at least 30 min to prevent bacterial carryover. Animals were then hand-picked either onto *lag-2* RNAi or empty L4440 vector control plates, cultured for 24 h at 16°C to the L4 stage, imaged, and background-subtracted fluorescence intensity was measured as above.

### Somatic gonad marker measurements

For measurement of somatic gonad cell number (Fig. S1G), the strain GS9692 expressing *arTi435(rps-27p::2xnls::gfp(flexon)::unc-54 3′UTR); arTi237(ckb-3p::Cre(opti)::tbb-2 3′UTR)* (Shaffer and Greenwald, 2022), which was specifically designed to activate enduring GFP expression in the cells of the somatic gonad, was used. Gonad cells were manually counted in this strain after inducing first-formed and starved plate dauers (Fig. S1F-H). We always observed the expected two DTCs and ten gonad blast cells in dauers of any age. Variation in expression levels/tissue depth sometimes complicated the detection of the cell at the center of the gonad primordium (more orange cells in Fig. S1G). Landmarks were selected based on relative position with DTCs at the gonad tips and the next-most-distal fluorescent cells being the SS cells used for measuring.

### Statistical analysis

All statistical analysis was performed using GraphPad Prism version 10.4.2 for Windows (www.graphpad.com). All tests and sample sizes are specified in the relevant figure legends.

Before any statistical tests were conducted on data, the data were first evaluated for normal distribution using the Shapiro–Wilk test for normality. Statistical tests were selected based on the appropriate test given the normality or non-normality of the data. If the data were normally distributed, Welch's *t*-test (two groups) or Welch's ANOVA (three or more groups) was used because it is more robust against Type I error rates than Student's *t*-test or an ordinary one-way ANOVA (Delacre et al., 2019; Derrick et al., 2016). When data violated the assumption of normality, non-parametric Mann–Whitney (two groups) or Kruskal–Wallis (three or more groups) tests were used. Post-hoc analyses for pairwise comparisons were selected based on appropriateness for the test used.

For curve fitting in Figs 3M, 4E, 5A and Fig. S2B,C, nonlinear regression models were compared to null-hypothesis linear regression models to test which was a better fit. Curves were fit to the means of the data at each time point. For exponential decay of LAG-2 protein, the plateau of the model was not constrained.

## Acknowledgements
We thank members of the Gordon laboratory and F.A.K. dissertation committee members Dr Rob Dowen and Dr Daniel Matute. for thoughtful comments on the manuscript. Some strains were provided by the CGC, which is funded by National Institutes of Health Office of Research Infrastructure Programs (P40 OD010440) and are to be requested directly from CGC.

## Competing interests
The authors declare no competing or financial interests.

## Author contributions
Conceptualization: F.A.K., K.L.G.; Formal analysis: F.A.K.; Funding acquisition: F.A.K., K.L.G.; Investigation: F.A.K., C.P.M., B.K.; Project administration: K.L.G.; Supervision: K.L.G.; Visualization: F.A.K.; Writing – original draft: F.A.K., K.L.G.; Writing – review & editing: F.A.K., C.P.M., K.L.G.

## Funding
Research reported in this publication was supported by the National Institute of General Medical Sciences (NIGMS) of the National Institutes of Health (R35GM147704 to K.L.G. and NIGMS Diversity Supplement 3R35GM147704-02S1 to support the work of F.A.K.). Support was also provided by the National Science Foundation (NSF CAREER 2442303). The content is solely the responsibility of the authors and does not necessarily represent the official views of the National Institutes of Health or the National Science Foundation. Open Access funding provided by the University of North Carolina at Chapel Hill. Deposited in PMC for immediate release.

## Data and resource availability
All relevant data and details of resources can be found within the article and its supplementary information.

## Peer review history
The peer review history is available online at https://journals.biologists.com/dev/lookup/doi/10.1242/dev.204972.reviewer-comments.pdf

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
