## [Peer Review File · Development (Cambridge, England)]

Pre-dauer starvation rapidly and reversibly reduces niche proliferative signaling to the *C. elegans* germ line

Fred A. Koitz, Camille P. Miller, Brian Kinney and Kacy Lynn Gordon

DOI: 10.1242/dev.204972

Editor: Mansi Srivastava

Review timeline

Original submission:	23 May 2025
Editorial decision:	17 July 2025
First revision received:	31 October 2025
Editorial decision:	1 December 2025
Second revision received:	22 December 2025
Accepted:	23 December 2025

Original submission

First decision letter

MS ID#: dev.204972

MS TITLE: Pre-dauer starvation decouples somatic from germline development with lifelong reproductive consequences in *C. elegans*

AUTHORS: Fred A. Koitz, Camille P. Miller and Kacy Lynn Gordon

Dear Dr Gordon,

I have now received all the referees' reports on the above manuscript, and have reached a decision. The referees' comments are appended below, or you can access them online: please go to:

As you will see, the referees express considerable interest in your work, but have some significant criticisms and recommend a substantial revision of your manuscript before we can consider publication. If you are able to revise the manuscript along the lines suggested, which may involve further experiments, I will be happy to receive a revised version of the manuscript. Your revised paper will be re-reviewed by one or more of the original referees, and acceptance of your manuscript will depend on your addressing satisfactorily the reviewers' major concerns. Please also note that Development will normally permit only one round of major revision. If it would be helpful, you are welcome to contact us to discuss your revision in greater detail. Please send us a point-by-point response indicating your plans for addressing the referees' comments, and we will look over this and provide further guidance.

Please attend to all of the reviewers' comments and ensure that you clearly highlight all changes made in the revised manuscript. Please avoid using 'Tracked changes' in Word files as these are lost in PDF conversion. I should be grateful if you would also provide a point-by-point response detailing how you have dealt with the points raised by the reviewers in the 'Response to Reviewers' box. If you do not agree with any of their criticisms or suggestions please explain clearly why this is so.

Reviewer 1

Advance summary and potential significance to field

In this manuscript, Koitz et al. investigate the early developmental differences during dauer diapause formation that lead to reproductive consequences in the postdauer adult. The authors use "progressive starvation" and constitutive dauer formation methods to analyze the gonad size and germ cell number of "early" and "late" dauers. They show that early dauers have larger gonads corresponding to greater germ cell numbers compared to late dauers. The authors also conclude that LAG-2 protein expression is decreased in the distal tip cell of both dauer types, but it rapidly increases again during dauer recovery. LAG-2 is the Notch ligand required for the germline stem cell niche, and its expression has previously been shown to be downregulated in starvation and high pheromone contexts in dauers and adults. Based on their results, the authors pose a model whereby longer starvation periods leading to dauer formation result in small gonads/fewer germ cells, correlating with a smaller brood size in adulthood.

Comments for the author

General comments about manuscript:

- 1) Some of the statements lack clarity throughout the manuscript. For example, lines 55-56 state that time spent in dauer does not affect lifespan, but line 58 states that dauers have increased lifespan. It is important to distinguish that duration of time in dauer does not affect lifespan, but any time in dauer (short versus extended time) results in increased mean lifespan or healthspan of the population (Hall et al 2010; Webster et al 2018). Similarly, on line 59, I think it is misleading to say that previous studies "disagree" about the effects of dauer on reproduction. The methods used in this list of citations to induce dauer vary greatly (daumone/starvation/crowding versus plates/liquid culture versus short/extended dauer), so a better conclusion would be that *C. elegans* dauers are sensitive to the conditions in which they are induced and maintained and adapt accordingly.
- 2) In the methods, the distinction between "early" and "late" dauers is not clear. Are all the dauers formed at the same time, and you are picking them for analysis early and late? Or, is it that you are picking dauers that have formed early and dauers that have been formed late? This method should be clarified in the methods and throughout because it has important implications for analysis of the results. The "early" dauer images in Figure 1 look more like L2ds or animals that were starved into dauer (which they were), and the late dauers look like dauers induced by crowding (thinner and dark). What are the methods used to ensure that you are picking dauers and not L2ds? There should also be wildtype images for the progressive starvation protocol in Figure 1.
- 3) The model proposed in this manuscript needs some additional discussion. The stated goal in the beginning is to examine differences in dauer gonads and how that may correlate with reproductive differences in postdauer adults. The authors state that more starvation = smaller gonad. But, do the late dauers, who are starved more than early dauers, shrink their gonads and have apoptosis of their germ cells? Or do the late dauers not undergo the last cell division as described in the manuscript? The authors need more discussion of their results and how the difference in gonads arise with respect to comment #2, especially since differences in lag-2 expression do not account for the phenotypes. If the late dauers are eating more than early dauers, would they not have larger gonads?
- 4) The authors state that smaller gonads lead to smaller brood sizes in postdauer adults. In Fig 1G, the *daf-2* animals have significantly smaller gonads than controls, but still exhibit the same brood size. How would this be explained? These results are hard to compare because of different graphs and controls. I also do not think using same control data for Fig 1A and 1C is appropriate for different conditions. Additionally, it would be helpful to the reader to label the graphs more specifically with the strains used instead of "control".
- 5) I think that the conclusions regarding lag-2 overexpression may be overstated given the data presented. The authors say that lag-2 protein increases within 2 hours of dauer recovery, but 3 of the 4 reporters used do not support this (Figure 5). The discrepancy is attributed to post-translational regulation of LAG-2 without data or a specific discussion of how that might happen. Lag-2 expression in these mutants in different conditions has been examined by multiple groups (including Pekar et al. and Karp and Greenwald 2013), so the authors should compare their results more explicitly with previous findings. Given that lag-2 expression is known to be regulated by TGF-beta signaling, it is a surprise to me that the authors did not continue to test *daf-7* mutants in these assays.

6) The results of the food removal experiment (Figure 2) are confusing and not convincing as is. How are L2ds continuing to feed for 24 hours without molting? I am convinced the early timepoints are still measuring L2d, but the late timepoint animals look larger and wider, suggesting they are experiencing additional larval molts. Is there an L2d marker that can be used to make sure animals are in the correct stage?

Specific points:

- 1) Line 118: Some differences in dauer germ cell number have been described based on early conditions (Ow et al 2018), so further clarity on the novelty of your study is needed here.
- 2) Line 138: "respectively" not needed here
- 3) Line 464: L1 arrest? Also, it is buried in the methods that animals are L1 arrested to synchronize. However, it is unclear whether this is used for all experiments for just some. Since L1 arrest can have major physiological consequences in *C. elegans*, the authors should be careful about their controls as it could cause confounding problems with dauer arrest.
- 4) When the authors hypothesize that the germline of dauers is notch-independent, are they referring to the last cell division before dauer formation is notch-independent or germ cell maintenance during dauer is notch-independent?

Reviewer 2

Advance summary and potential significance to field

The dauer larva of *C. elegans* is a fascinating, facultative state of diapause in response to difficult environmental conditions. This ms. by Koitz et al. describes the anatomy of the somatic gonad and germline under different dauer-inducing conditions. The descriptive work is carefully done but, in my opinion, does not represent "a significant and novel contribution to our understanding of developmental mechanisms" and thus is of insufficient general interest.

Major concerns:

(1) throughout, the smaller number of germ cells and broods after recovery when dauers are formed using the progressive crowding+starvation protocol than in the single DAF-2 Insulin Receptor mutant is interpreted as indicating that "the fed state itself, rather than nutrient sensing via IIS, is regulates pre-dauer germline growth." I did not understand the reasoning because both starvation and crowding reduce both TGF β and insulin-like proteins in sensory neurons, i.e. the two signals are not uniquely regulated by different inputs (several papers referenced in Baugh and Hu, Genetics 2020). Furthermore, the authors do not examine the gonad phenotypes when both the TGF β and DAF-2 inputs are compromised (or just the TGF β input is compromised), raising the possibility that this partial redundancy may account for the different dauer gonad phenotypes.

(2) The dauer germline is maintained in a Notch-independent, quiescent state. (section beginning with line 290-) Narbonne and Roy (Development 2006) already provided evidence that the dauer germline remains quiescent independent of GLP-1/Notch (although they did not show all the data). Apart from that, the critical question would seem to be the nature of the Notch-independent quiescent state and that does not seem to be addressed.

(3) I found the reasoning of the later Results sections, 322-end, hard to follow. I am not persuaded by what I read. Downregulation of LAG-2/ligand in the Distal Tip Cells (DTCs), the germline niche, in L4 larvae response to adverse environmental conditions has already been established (Pekar et al., Development 2017) in some molecular detail at the level of transcription, so the reduction of LAG-2 in the DTCs in L2d is not in itself novel. What appears to be novel is the argument that the regulation is occurring post-translationally. However, the experiments giving rise to this conclusion have caveats not addressed. While it is commendable that the authors addressed caveats that might have been serious if they had only relied on a highly artificial transcriptional lag-2p multicopy transgene reporter by looking at a polycistronic T2A knock-in expressing a co-regulated fluorescent histone, it was unclear to me how perdurance of the histone was accounted for; this would seem to be an important caveat of this approach. I am unconvinced that the level of regulation is at the level of protein trafficking or stability, which would seem to require expressing fluorescent LAG-2 from heterologous sequences.

(4) I also did not understand the contention that somatic and germline development are "decoupled" if the mechanism includes modulation of the stemness cue and an independent imposition of both somatic and germline quiescence. Perhaps I'm not understanding what evidence there is that the imposition of quiescence is different for the somatic gonad and germline--they state that Cassada and Russell (1975) showed that the germline division stopped before somatic development but to me, that is not really decoupling, it is that they both stop at the tissue-specific appropriate checkpoint.

Reviewer 3

Advance summary and potential significance to field

The manuscript submitted by Koitz et al. comprises a string of interesting observations that link various starvation regimens to the growth and proliferation of the gonad, and more particularly, to the germ line. The authors claim these regimens result in an uncoupling between the somatic gonad and germ line growth/division program, while also delineating several distinct means of interrogating the various signals and their effects on the intrinsic growth programs that drive the development of the germ line during the dauer diapause stage. In a second part of the manuscript the authors then focus their attention to how nutritional status and these various feeding/starvation regimens affect the levels of the Notch ligand LAG-2 in the dauer larva. Their final experiments drive the authors to conclude that the germ cells are held in quiescence in a Notch independent manner during the dauer stage. Although the observations are well described and quantified, I find that some of the authors' conclusions are not particularly well founded and would require significantly more experiments to corroborate their claims.

Comments for the author

The work that is submitted is indeed interesting, but the rationale, the presentation, and the description/discussion of their various feeding regimens is quite confusing at times, while it often lacks any discussion of biological or physiological relevance. Perhaps more disappointing is the lack of any attempt to better understand these phenomena with the available genetic or cell/developmental biological tools. As a result, the manuscript reads as a list of interesting observations that are largely descriptive, without really providing any further insight as to how these phenomena are regulated or what their biological significance might be.

The designs of the experiments are generally straightforward (albeit sometimes difficult to follow). Most of the observations are quantitatively sound and the data are acquired with care and attention. As such, I find them mostly convincing, just complicated and confusing without reason. I have listed a number of points that I believe are important for the authors to address in order to reinforce their conclusions.

Lines 237-240 they describe an uncoupling of the soma and the germ line in these various feeding regimens. I think that there are other examples where the development of the soma and the germ line are uncoupled during various stresses, including dauer, but since the authors are convinced of this uncoupling in this context, it would be advantageous to show the evidence that the somatic cells are not affected by these feeding regimens without simply referencing Cassada. That might include comparing cell cycle timing etc...to demonstrate that the soma is not equally slowed and actually follows the germ cells.

The L1 diapause and the dauer diapause are not identical and the genes that are affected are not the same, although there may be some overlap. Although the germ cells present in the dauer larva do resemble L1 PGCs in their compacted state, this has not been described in dauer in any significant detail. The authors could use H3K9me3 or HPL-1 staining to evaluate if there are differences between the various regimens and between dauer larvae.

Lines 257-263 This is an interesting observation, but is only superficially supported by the work shown here. I think that to make this conclusion, much more work must be completed to compare somatic growth development and its scaling to the constraints imposed by the gonad or the germ cell population.

Line 292- "the" LAG-2 ligand

Germ line is a noun; germline is an adjective. Wild type is a noun; wild-type is an adjective...throughout

Lines 300-303 The text indicates that L1 PGCs are arrested in a notch independent manner is not presented, while no citation is provided.

Lines 320-321 This is a bit contentious. There is no evidence that this is fully Notch independent. That the germ cells do not go into meiosis or exhibit meiotic features such as HIM-3 expression (was not shown) it would indicate that Notch is still blocking cells from engaging the full meiotic program and therefore must be functional. Perhaps there is another means of assessing Notch function at the molecular level to indicate that indeed it is extinguished during this state.

Lines 330-332 This observation suggests that the differences in the number of germ cells that are present are not a direct readout of LAG-2/GLP-1 signalling, but rather that LAG-2 is probably necessary, but is not the main constraint that dictates whether the cells proliferate or not.

Line 354- insert that "the" dauer germ line...

Lines 441-445 Somatic gonad development clearly responds to dauer and/or starvation cues as it is much more compact. Individual cell size was not measured but there is clearly an adaptive scaling that takes place. So although the cell cycle uncoupling may take place as indicated by the authors. The germ line is under exquisite regulatory control, perhaps mostly by AMPK and PTEN, which largely adjusts growth and division to reflect the challenge, while presumably doing little to nothing to alter growth or division in the somatic gonad or the soma in general. This somatic regulation would stem from other cues that may ultimately impinge on one of the CDK inhibitors, which have significant effects on the somatic tissues.

The authors use their imaging data to suggest that LAG-2 is under post-translational regulation to reduce its activity/abundance during the dauer stage. This should be elaborated in much greater detail, ideally by identifying the regulator and/or the stage-specific modification on the polypeptide itself by mass spectrometry.

Line 906 add a with after compared

It is unclear why the progressive starvation experiment was performed with a mixed population, whereas synchronized populations were used in the other described experiments. Would a synchronized population not make more sense for the starvation experiment, as you could ensure all animals were starved for the same duration? Perhaps the authors could clarify why they used a different population.

I am surprised the authors did not repeat at least part of the experiments in Fig 2 with the *daf-7* mutant. The data would be even more compelling to show that the potential decoupling they have revealed occurs regardless of the modality of dauer induction.

Fig 5A-B: I wonder if the authors could employ a qPCR method to directly measure levels of LAG-2 mRNA in a more quantitative manner. It would be more accurate than simply looking at a transcriptional reporter and would nicely complement their other data.

The image resolution of some of the figures seem low. In Fig 1 for example, some of the graph elements like the axes and points are much blurrier than the labels.

First revision

Author response to reviewers' comments

Dear Dr. Srivastava and three anonymous Reviewers,

Thank you for the opportunity to revise our manuscript. We have reframed and focused the paper and have clarified our descriptions of our experiments, results, and interpretations, to address concerns about how our work relates to what was previously known. We conducted extensive experiments on a mutant strain, *daf-7(e1372)*, which was the main experimental suggestion made by two Reviewers. In all instances, they support our conclusion that we observe a novel mode of nutritional regulation of LAG-2 in the DTC during dauer entry and recovery, distinct from the *daf-7*-dependent regulation of *lag-2* transcriptional reporters that was previously observed in later stage larval animals by Pekar et al., 2017. Indeed, neither *daf-2* nor *daf-7* are required for the starvation-induced germline stem cell niche depletion of LAG-2 protein nor the germline response to food removal. We added finer time courses of the LAG-2 protein drop upon removal from food in both mutant genotypes (Fig. 3) and of the post-dauer recovery of LAG-2 and our three transcriptional reporters in the DTC (Fig. 5). We have also created five supplemental figures with additional data, controls, and statistical analyses. We believe the revision is more tightly argued and comprehensively supported thanks to the Reviewers' feedback.

We reframed the paper as a test of reproductive system differences in dauers that are induced by different methods, allowing us to dispense with the "early" vs. "late" dauer framework that the reviewers found problematic. We added data obtained using published methods for inducing dauer by starvation (adding the Baugh liquid culture protocol to the starvation-on-plate protocol we originally used) and crowding (Hall high density plating protocol added to the first-formed dauers we originally examined). The predicted patterns are observed in all cases: animals that feed more before dauer have bigger gonads with more germ cells and better reproductive recovery than animals that entered dauer in abject starvation. We have focused on the germline stem cell niche and germline, and de-emphasized the argument about decoupling of somatic and germ line development that did not resonate with reviewers (we have changed the title accordingly).

In several comments, the Reviewers questioned the novelty of our work by referencing previously published studies. In each case, we respectfully rebut these claims with directly quoted text and figures from those prior studies in our point-by-point response. We have done our best to scour the literature for the claims made by the Reviewers, but it is of course possible that we missed papers to which the Reviewers refer and would welcome their suggestions. However, we ask that our findings not be dismissed out of hand as lacking novelty based only on the Reviewers' assertions, without a similar level of specificity to that which we provide for our sources, below, in the point-by-point.

We believe that one of the strengths of our paper is that some of our discoveries "feel" like classic findings that could have been reported decades ago, yet an attentive review of the literature shows that these features of the dauer gonad—the existence of dauer germline size differences, feeding-dependent pre-dauer germline growth, and germline stem cell Notch-independence in dauer—are previously untested and undescribed. Adding to these "old school" experiments, we use quantitative image analysis of endogenously tagged LAG-2 protein and several *lag-2* transcriptional reporters. Altogether, we discovered that regulation of the stemness cue in the germline stem cell niche is highly responsive to environmental conditions independent of two major nutrient sensing pathways, mediated at the protein level, and is in fact one of the earliest features of dauer recovery. This work makes a substantial contribution to our understanding of environmental regulation of stem cell niche signaling.

With gratitude for your consideration on behalf of my coauthors,

Kacy Gordon

Reviewer 1

SUMMARY OF THE ADVANCE MADE IN THIS PAPER AND ITS POTENTIAL SIGNIFICANCE TO THE FIELD

In this manuscript, Koitz et al. investigate the early developmental differences during dauer diapause formation that lead to reproductive consequences in the postdauer adult. The authors use "progressive starvation" and constitutive dauer formation methods to analyze the gonad size and germ cell number of "early" and "late" dauers. They show that early dauers have larger

gonads corresponding to greater germ cell numbers compared to late dauers. The authors also conclude that LAG-2 protein expression is decreased in the distal tip cell of both dauer types, but it rapidly increases again during dauer recovery. LAG-2 is the Notch ligand required for the germline stem cell niche, and its expression has previously been shown to be downregulated in starvation and high pheromone contexts in dauers and adults. Based on their results, the authors pose a model whereby longer starvation periods leading to dauer formation result in small gonads/fewer germ cells, correlating with a smaller brood size in adulthood.

We disagree with the underlined statement. The downregulation of LAG-2 protein in the niche under starvation has not been reported at any stage. Transcriptional downregulation of *lag-2* reporters in the niche has been reported under low food and high pheromone conditions only in L4 stage worms, not dauers or adults (Pekar et al. 2017):

“We built DTC membrane-bound reporters (using GFP or mCherry fusions to the PH domain of the rat PLC1δ1) driven by the 3 kb *lag-2* upstream region used by others (Henderson et al., 1994). We introduced the new reporter transgenes into the worm genome using microparticle bombardment, a technique that results in fewer copies than traditional transgenes borne on extrachromosomal arrays. We examined the DTC in the fourth larval stage (L4) as DAF-7/TGFβ signaling affects proliferative germ cell accumulation before and during this stage (Dalfó et al., 2012).”

NOTE: Figure provided for reviewer has been removed. It showed Figure 5 from Park, D., Estevez, A., Riddle, D.L. (2010) Antagonistic Smad transcription factors control the dauer/non-dauer switch in *C. elegans*. *Development* 137, 477-485. doi: <https://doi.org/10.1242/dev.043752>. We have removed unpublished data that had been provided for the referees in confidence.

Park et al. (2010) find that *lag-2* transgene expression is negatively regulated by *daf-8* in the adult DTC, and discuss, but do not test the potential relevance of *lag-2* suppression to the role of this pathway in dauer:

Park et al., 2010, Figure 5 reflects the study's focus on *lag-2* in the non-dauer only.

In dauers, Narbonne and Roy 2006 report “strong” expression of several *lag-2* reporters in dauer DTCs, but no quantitative comparison was made:

“Paradoxically, we noticed that the expression of the Notch ligand remains strong in the DTCs during the dauer stage (Fig. 1A,B)...”

and in the legend of Narbonne & Roy 2006 Figure 1:

“(B) An integrated *lag-2p::GFP* transgene(*qls56*) is strongly expressed in the DTCs throughout the dauer stage. A *lag-2p::CFP* transgene (*arEx645*) containing the full 6.2 kb promoter (Chen and Greenwald,2004) was also strongly expressed in the DTCs of dauer larvae (data not shown). We similarly found strong GFP expression in the DTCs of *qEx308* dauer larvae (data not shown), which contain a functional *lag-2p::LAG-2::GFP lag-2(q411)* rescuing construct (S. Crittenden and J. Kimble, personal communication).”

We would appreciate suggestions from the Reviewer.

SUGGESTIONS TO AUTHORS

General comments about manuscript:

1) Some of the statements lack clarity throughout the manuscript. For example, lines 55-56 state that time spent in dauer does not affect lifespan, but line 58 states that dauers have increased lifespan. It is important to distinguish that duration of time in dauer does not affect lifespan, but any time in dauer (short versus extended time) results in increased mean lifespan or healthspan of the population (Hall et al 2010; Webster et al 2018).

We have ensured that it is always clear when we are talking about total lifespan vs. adult lifespan. We have added the Hall et al., 2010 reference; we had meant to include it. Thank you for pointing it out.

Similarly, on line 59, I think it is misleading to say that previous studies "disagree" about the effects of dauer on reproduction. The methods used in this list of citations to induce dauer vary greatly (daumone/starvation/crowding versus plates/liquid culture versus short/extended dauer), so a better conclusion would be that *C. elegans* dauers are sensitive to the conditions in which they are induced and maintained and adapt accordingly.

We appreciate this feedback and changed our description of the relationship between time in dauer and later reproduction from one of "disagreement" in the field to one that is, as the Reviewer states, sensitive to experimental conditions. The text (lines 57-60) has been revised to read:

Previous studies discovered different long-term effects on post-dauer reproduction depending on the dauer-inducing conditions worms experienced (Klass and Hirsh 1976; Kim and Paik 2008; Hall et al. 2010; Ow et al. 2018; Webster et al. 2018), focused on recovery.

2) In the methods, the distinction between "early" and "late" dauers is not clear. Are all the dauers formed at the same time, and you are picking them for analysis early and late? Or, is it that you are picking dauers that have formed early and dauers that have been formed late? This method should be clarified in the methods and throughout because it has important implications for analysis of the results.

The "early" dauer images in Figure 1 look more like L2ds or animals that were starved into dauer (which they were), and the late dauers look like dauers induced by crowding (thinner and dark). What are the methods used to ensure that you are picking dauers and not L2ds? There should also be wildtype images for the progressive starvation protocol in Figure 1.

We revised the "early" vs. "late" dauer paradigm. We now compare "crowded" vs. "starved" dauers using established protocols to induce dauer under those conditions. Dauers produced by these protocols were all isolated by SDS treatment, which kills worms without a dauer cuticle, including L2d worms. The animals we studied here all have visible alae and are unambiguously dauers. We have added Fig. S2A-A' showing alae as a diagnosable feature of dauer.

We made bigger restructuring changes to Fig. 1 to include otherwise wild type worms with fluorescent markers as the first-displayed subject for each dauer-induction protocol, with the exception of Constitutive dauers, since wildtype worms do not constitutively form dauers.

3) The model proposed in this manuscript needs some additional discussion. The stated goal in the beginning is to examine differences in dauer gonads and how that may correlate with reproductive differences in postdauer adults. The authors state that more starvation = smaller gonad. But, do the late dauers, who are starved more than early dauers, shrink their gonads and have apoptosis of their germ cells? Or do the late dauers not undergo the last cell division as described in the manuscript?

The authors need more discussion of their results and how the difference in gonads arise with respect to comment #2, especially since differences in lag-2 expression do not account for the phenotypes.

Our model is that abject starvation before dauer causes reduction in DTC LAG-2 (Fig. 3) and simultaneous cessation of germ cell proliferation (Fig. 2). Less germ cell proliferation before dauer leads to fewer germ cells and smaller gonads in dauer (shown in the food removal experiments in Fig. 2). We also show that LAG-2::mNG expression is dynamically restored during dauer recovery (Fig. 5), anticipating shifting requirements for Notch signaling to maintain germline stem cells and the resumption of germ cell cycling (Fig. 4). We summarize in the Discussion on page 11:

We propose a model by which steady but fairly low levels of *lag-2* transcription and translation in dauer are coupled with LAG-2 protein degradation triggered by starvation. Upon dauer recovery, protein degradation ceases, leading to the rapid accumulation of LAG-2 protein in the DTC.

We have rigorously ruled out germline apoptosis as causing small gonads. We did not include these negative results in the initial manuscript, as this seemed to be a sidetrack and our manuscript is already pushing the word limit, but we can easily furnish them if required.

If the late dauers are eating more than early dauers, would they not have larger gonads?

In our stated model, late dauers eat less than early dauers. Early dauers form first on crowded plates while there is still food to be eaten in L2d. Late dauers form later, after food has been exhausted. We have pulled the early/late paradigm from this paper, and describe these “first formed” dauers in the Methods on page 15:

Recovery of first-formed dauers

As above, evaluated <4 days after food exhaustion was noted, before a robust population of dauers are visible on the plate. Dauers were isolated with a treatment of 1% SDS for 20 minutes (see Dauer isolation, below).

4) The authors state that smaller gonads lead to smaller brood sizes in postdauer adults. In Fig 1G, the *daf-2* animals have significantly smaller gonads than controls, but still exhibit the same brood size. How would this be explained? These results are hard to compare because of different graphs and controls. I also do not think using same control data for Fig 1A and 1C is appropriate for different conditions. Additionally, it would be helpful to the reader to label the graphs more specifically with the strains used instead of "control".

We thank the reviewer for this suggestion. We now ensure that genotype and treatment labels are clear and thorough in the figures and legends in our substantially revised Figure 1. The control originally shown in Fig. 1G (Revised Fig. 1F) is a never-dauered control of the *daf-2(e1370)* genotype, not a wild-type control. This figure shows that if *daf-2(e1370)* mutants enter dauer constitutively at high temperature, upon their recovery they have a normal brood compared to the never-dauered *daf-2(e1370)* reared at the lower temperature. The same is true for facultative *daf-2(e1370)* “first-formed” dauers reared at the lower temperature, but *not* for the “starved” dauers reared at the low temperature which have smaller gonads and broods. These data now appear in Fig. 1C, D, and F.

We have made substantial changes to Figure 1, improving its clarity. Now we group by genotype with otherwise wildtype worms with fluorescent markers shown in the first part of the figure, and the *daf-c* mutants shown in separate panels, instead of grouping by facultative vs. constitutive dauer. Only *daf-c* mutants can form constitutive dauers, the wild type can only form facultative dauers, so there is no way to have a constitutive, wild-type dauer.

5) I think that the conclusions regarding *lag-2* overexpression may be overstated given the data presented. The authors say that *lag-2* protein increases within 2 hours of dauer recovery, but 3 of the 4 reporters used do not support this (Figure 5). The discrepancy is attributed to post-translational regulation of LAG-2 without data or a specific discussion of how that might happen.

We thank the reviewer for bringing to our attention the need to more clearly describe the reporters in Fig. 5. Only one reagent reports LAG-2 protein (*lag-2(cp193[lag-2::mNeonGreen^{3xFlag}])*). Another is a polycistronic tagged histone knock in at the endogenous locus of *lag-2*, in which tagged histone is translationally cleaved from LAG-2 (*lag-2(bmd202[lag-2::P2A::H2B::mTurquoise2^{lox5111}^2xHA])*). Two others are established transcriptional reporters that make membrane-localized fluorescent proteins from a multicopy integrated array transgene (*qls154(lag-2p::myr::tdTomato)*) and a single-copy integrated transgene *cpls122(lag-2p::mNeonGreen::PLCδ1^{PH})* both driven by a ~3kb upstream *lag-2* promoter sequence. These are all previously published transgenes, and we cite sources and describe them appropriately in the Methods. For a summary, see table below.

The *differences* among these reporters implicate protein-based regulation of LAG-2 recovery with evidence of unchanged *lag-2* transcriptional output on the same timeline (see next). The data in Fig. 5 demonstrate that the endogenously tagged protein is regulated differently than the transcriptional reporters, including a transcriptional reporter made from a polycistronic transcript that only physically diverges from native LAG-2 after the proteins are synthesized. We have added Fig. S5 showing that the fluorescent histone does not have a long half-life (see response to Reviewer 2). In our revision, we have added a discussion of the types of post-translational regulation that might affect LAG-2.

Table of *lag-2* reporters, color-coded by false color in Fig. 5

Transgenic element	Construct	Recapitulates lag-2 txn?	Recapitulates lag-2 post-txn control?	Recapitulates LAG-2 protein dynamics?	Protein/cell structure labeled?	Source
lag-2(cp193[lag-2::mNG^{3xFLAG}])	CRISPR/Cas9-mediated knock-in of mNeonGreen to endogenous lag-2 , C-terminal	Maximally	Maximally	Maximally	Functional LAG-2 made under endogenous regulatory control, punctate at cell membrane	Gordon et al., 2019
lag-2(bmd202[lag-2::P2A::H2B::mTurquoise2¹¹¹2xHA])	CRISPR/Cas9-mediated knock-in of P2A self-cleaving peptide, histone coding sequence, and mTurquoise2 at endogenous lag-2 locus	Maximally	Maximally	Protein production only	Fluorescent histone 2B, nuclear	Medwigin-Kinney et al., 2022
cpls122(lag-2p::mNeonGreen::PLCδ1^{PH})	~3kb noncoding sequence upstream of lag-2 driving expression of mNeonGreen with PH domain from an integrated single copy transgene in the Mosci site on Ch. II	Partially	No, no lag-2 transcript included, uses let-858 terminator	No	mNeonGreen localized to plasma membrane, even labeling of plasma membrane	Linden et al., 2017
qls154(lag-2p::myr::tdTomato)	~3kb noncoding sequence upstream of lag-2 driving expression of myristoylated TdTomato from an integrated multicopy transgene	Partially	No, no lag-2 transcript included, uses unc-54 3' UTR	No	TdTomato localized to plasma membrane, even labeling of plasma membrane	Byrd et al., 2014

Lag-2 expression in these mutants in different conditions has been examined by multiple groups (including Pekar et al. and Karp and Greenwald 2013), so the authors should compare their results more explicitly with previous findings.

Karp and Greenwald, 2013 test how different *lag-2* promoter deletion constructs do or do not have expression in the DTCs among other cells, but they focus on different cells of the proximal gonad (the AC and VPCs). We added this useful reference to our description of differences among the transgenes (above) and of how *lag-2* plays multiple roles in multiple cell types in dauer worms.

We cite Pekar et al., 2017 copiously. We have added a more explicit comparison between our results and those of Pekar et al (underlined below). In brief, Pekar's study looked only at transcriptional regulation of *lag-2* reporters, never at LAG-2 protein, and only examined L4 larvae (direct source text for this is cited in our first reply to Rev. 1, above). Our new Fig. S3 reveals some previously unidentified features of the regulation of *lag-2* by *daf-7*.

When study *lag-2* expression in *daf-7(e1372)* mutants, we reference that work (on page 7):

Comparing otherwise wild-type worms to *daf-2(e1370)* and *daf-7(e1372)* reveals that both IIS and TGF- β are required for normal expression levels of *lag-2* promoter activity and LAG-2::mNG protein in the DTC in fed L2 animals (Fig. 3E-I); this concurs with what is known about the positive transcriptional regulation of the *lag-2* promoter by *daf-7* later in continuous, non-dauer development (Pekar et al. 2017).

...

Endogenous LAG-2::mNG protein abundance is highly responsive to the onset of starvation in L2d in both genotypes (Fig. 3K-M). LAG-2::mNG drops from the L2d 20h fed baseline within 1 hour of food removal in both *daf-2(e1370)* (-27.7%) and *daf-7(e1372)* (-48.1%) mutants, and continues to fall in an exponential decay (Fig. 3M).

...

This result distinguishes the process observed here from the *daf-7*-dependent downregulation of *lag-2* transcriptional reporters observed in L4 animals reared in unfavorable conditions (Pekar et al. 2017; Dalfó et al. 2012). Further comparisons of *lag-2* promoter activity and protein abundance in *daf-7(e1372)* mutants reinforced our conclusions that the role of *daf-7* in regulating *lag-2* in pre-dauer and dauer stages (Fig. S3) is distinct from the simple positive relationship reported during continuous development by Pekar et al. (2017).

Given that *lag-2* expression is known to be regulated by TGF-beta signaling, it is a surprise to me that the authors did not continue to test *daf-7* mutants in these assays.

We have added extensive analyses of *daf-7(e1372)* mutants to the paper (Figs. 1-3, Fig. S2, S3).

6) The results of the food removal experiment (Figure 2) are confusing and not convincing as is. How are L2ds continuing to feed for 24 hours without molting?

This experiment was done with *daf-2(e1370)* mutants, which we label in the Fig. 2 legend, Methods, and the Results text describing the figure. We have now added this genotype label to the figure image itself.

Constitutive dauer entry is known to take about 80 hours in continuously fed *daf-2(e1370)* mutants, as is reported by Karp's Wormbook Chapter "Working with dauer larvae" (emphasis mine):

3.5.4. Protocol for dauer induction by *Daf-c* mutations

1. Allow gravid adult hermaphrodites to lay eggs for several hours on standard plates* (Maintenance of *C. elegans*)
2. Remove adults if a synchronous population is desired.
3. Incubate progeny at 25 °C to induce dauer formation.
4. Dauer larvae will form by approximately 48 hours at 25 °C for *daf-7(e1372)*† and by **approximately 80 hours at 25 °C for *daf-2(e1370)*.**

NOTE: Figure provided for reviewer has been removed. It showed Table 2 from Gems, D., Sutton, A.J., Sundermeyer, M.L., Albert, P.S., King, K.V., Edgley, M.L., Larsen, P.L., Riddle, D.L. (1998) Two pleiotropic classes of *daf-2* mutation affect larval arrest, adult behavior, reproduction and longevity in *Caenorhabditis elegans*. Gen 150, 129-155. doi: <https://doi.org/10.1093/genetics/150.1.129>. We have removed unpublished data that had been provided for the referees in confidence.

It is established that 100% of *daf-2(e1370)* mutants will enter dauer or arrest prior to dauer (see ^d, right) when raised at the restrictive temperature (Gems et al., 1998, see Table 2 at right), despite feeding:

"Culture methods and strains: Animals were maintained monoxenically in 60-mm Petri dishes containing 10 ml NG agar seeded with *Escherichia coli* OP50 as the food source (Brenner 1974)." Gems et al., 1998

Therefore, these L2ds can feed for 24 h without molting because the animals have a mutant genotype and are being reared at a temperature at which 100% of them will molt into dauer or

arrest before dauer, and that molt takes a long time compared to the normal L2 larval stage of continuously fed animals. As we explain in our paper on page 5:

Using *daf-c* mutants at 25°C ensured that we could vary food exposure while keeping every animal in the L2d dauer entry program. Worms were reared on plates with bacterial food for the specified number of hours at 25°C (Fig. 2A), removed from food at 25°C, and maintained until a population of majority dauers was noted (see Methods).

We removed worms from food during the L2d stage at 12, 16, 20, 24 hours, and allowed others to feed continuously until dauer (48+ hours (Karp 2018)).

We illustrate in our schematic Fig. 2A and with data in Fig. 2C and D that the animals that feed more during L2d then enter dauer more quickly than the worms that have been removed from food after shorter durations of feeding. Our findings are in line with those of Cassada and Russell, 1975, who showed that dauer formation took longer when worms were grown on lower bacterial food titres.

We have further improved the way we describe our experimental design in the Results and Methods. To this food removal assay we have added a parallel one for *daf-7(e1372)* (Fig. 2C and Fig. S2).

I am convinced the early timepoints are still measuring L2d, but the late timepoint animals look larger and wider, suggesting they are experiencing additional larval molts. Is there an L2d marker that can be used to make sure animals are in the correct stage?

As explained above, these are *daf-2(e1370)* mutants. At 25°C, this strain will not undergo additional molts beyond dauer (see Gems et al., 1998, Table 2 quoted above).

The animals that fed for longer durations (labeled 20 h, 24 h, Continuously fed) are larger because they are developmentally older and have eaten more food. All of the left column of worms were assayed during the L2d period, which can be detected by lack of cuticular alae, lack of buccal plug, lack of a proximal somatic gonad separating the two arms of the germline, and of course by the *daf-c* mutant phenotype, which ensure that no molts beyond the dauer molt will take place in these rearing conditions. This satisfactorily validates these are L2d worms, and unfortunately, as Karp's Wormbook chapter "Working with dauer larvae" states: "No L2d-specific fluorescent markers have been characterized."

Specific points:

1) Line 118: Some differences in dauer germ cell number have been described based on early conditions (Ow et al 2018), so further clarity on the novelty of your study is needed here.

NOTE: Figure provided for reviewer has been removed. It showed part of Figure 6 from Ow, M.C., Borziak, K., Nichitean, A.M., Dorus, S., Hall, S.E. (2018) Early experiences mediate distinct adult gene expression and reproductive programs in *Caenorhabditis elegans*. PLoS Genet. 14, e1007219. doi: <https://doi.org/10.1371/journal.pgen.1007219>. We have removed unpublished data that had been provided for the referees in confidence.

Ow et al. 2018 did not measure germ cell number in the dauer stage itself, but only in recovery. That study measured germ cell number in postdauer animals with vulval morphologies reflecting L3, L4.1, and L4.4:

"To compare developmentally synchronized animals, we examined GFP levels of postdauer and control animals that experienced either pheromone or starvation conditions and exhibited the vulva morphology characteristic of L3, L4.1, and L4.4 larval animals [47, 48], at which times the mitotic and transition zones are evident..."

See stages described on the lower axes of Figure 6 of that paper (above right).

We reference these results in relation to our findings on page 12:

Thus we interpret the observation that germ cell number during recovery from dauer differs depending on how dauer was induced (Ow et al. 2018) to reflect differences in initial dauer germ cell number induced by these same treatments.

2) Line 138: "respectively" not needed here

We have reworded this sentence.

3) Line 464: L1 arrest?

We found that L1 arrested germlines were Notch independent, previously in lines 300-303. We have removed these data from the paper since they were deemed to be tangential.

Also, it is buried in the methods that animals are L1 arrested to synchronize. However, it is unclear whether this is used for all experiments for just some. Since L1 arrest can have major physiological consequences in *C. elegans*, the authors should be careful about their controls as it could cause confounding problems with dauer arrest.

Word limits prevent us from describing all experimental protocols in the results, but we have ensured that our Methods describe when synchronization via L1 arrest was used, which was only in *daf-c* food removal experiments, and for initiating the liquid culture protocol we added to Fig. 1, according to Hibschan et al., 2021. .

Perhaps the Reviewer alludes to the fact that passing through L1 arrest makes it less likely that the same animals will later enter dauer (see Karp's Wormbook chapter "Working with dauer"). We find no evidence that this is the case for *daf-c* animals of *daf-2(e1370)* or *daf-7(e1372)* genotypes, the genotypes for which we used L1 arrest to synchronize worm populations.

4) When the authors hypothesize that the germline of dauers is notch-independent, are they referring to the last cell division before dauer formation is notch-independent or germ cell maintenance during dauer is notch-independent?

The experiment in Fig. 4A-B demonstrates that the germ line in dauer (not the terminal pre-dauer cell division) is maintained in a Notch-independent manner, as 100% of *glp-1(bn18)* dauers recover as fertile adults after 24 h at the restrictive temperature. This is a functional demonstration of Notch-independence, not a hypothesis.

Reviewer 2**SUMMARY OF THE ADVANCE MADE IN THIS PAPER AND ITS POTENTIAL SIGNIFICANCE TO THE FIELD**

The dauer larva of *C. elegans* is a fascinating, facultative state of diapause in response to difficult environmental conditions. This ms. by Koitz et al. describes the anatomy of the somatic gonad and germline under different dauer-inducing conditions. The descriptive work is carefully done but, in my opinion, does not represent "a significant and novel contribution to our understanding of developmental mechanisms" and thus is of insufficient general interest.

We appreciate that the Reviewer recognizes that our work was carefully done, and we look forward to the opportunity to bring attention to the novelty of our findings in response to specific concerns below.

Major concerns:

(1) throughout, the smaller number of germ cells and broods after recovery when dauers are formed using the progressive crowding+starvation protocol than in the single DAF-2 Insulin Receptor mutant is interpreted as indicating that "the fed state itself, rather than nutrient sensing via IIS, is regulates pre-dauer germline growth." I did not understand the reasoning because both starvation and crowding reduce both TGF β and insulin-like proteins in sensory neurons, i.e. the two signals are not uniquely regulated by different inputs (several papers referenced in Baugh and Hu, Genetics 2020).

First we must clarify our results. While *daf-2(e1370)* mutants that are reared at 25°C on food have a large number of germ cells, the same mutant subjected to pre-dauer starvation has few germ cells. The same is true for *daf-7(e1372)*.

Thus we find that neither *daf-2* nor *daf-7* is required for pre-dauer starvation to yield a smaller dauer gonad or pre-dauer feeding to yield a larger dauer gonad. Our question here is not about

inputs to the pathways, but whether the pathways are necessary for our phenotype, which we investigate with a simple genetic analysis. Since the phenotype (of feeding-dependent germline growth) persists in both mutant genotypes, we conclude the normal function of those genes is not required for the phenotype.

We appreciate the complex interplay of IIS and TGF β signaling in nutrient sensing and dauer entry, and added the useful references suggested by the reviewer.

Furthermore, the authors do not examine the gonad phenotypes when both the TGF β and DAF-2 inputs are compromised (or just the TGF β input is compromised), raising the possibility that this partial redundancy may account for the different dauer gonad phenotypes.

We appreciate the suggestion to extend our analysis of *daf-7(e1372)* mutants. In this revision, we added analyses of the effect of *daf-7(e1372)/TGF β* mutation on germline growth in L2d food-removal experiments (Fig. 2 and Fig. S2) and on LAG-2 protein and reporter expression (Fig. 3 and Fig. S3). With this additional data, we can more confidently say that nutritionally-sensitive arrest of germ cell proliferation and reduction in LAG-2 occur even in the *daf-7(e1372)* mutant and are therefore regulated in a *daf-7*-independent manner. We find no record of *daf-2(e1370); daf-7(e1372)* double mutants in the extensive dauer genetics literature or at the CGC.

(2) The dauer germline is maintained in a Notch-independent, quiescent state. (section beginning with line 290-) Narbonne and Roy (Development 2006) already provided evidence that the dauer germline remains quiescent independent of GLP-1/Notch (although they did not show all the data).

Narbonne and Roy (2006) show that when *daf-2(e1370)* mutant germ cells lose *glp-1/Notch* signaling as young larvae *before entering dauer*, all the germ cells will enter meiotic pachytene (see panel G below), but they do not go on to form sperm while in dauer. (While the timing is not described in detail, we know that Notch signaling is lost in young, pre-dauer larvae in this experiment because the elevated temperature at which *daf-2(e1370)* worms must be reared from L1 to form dauer constitutively is the restrictive temperature for *glp-1(e2141)*.) This experiment shows quiescence with respect to meiotic progression and gamete differentiation *even in the absence of Notch* (which typically inhibits meiotic entry). This is not a test of Notch-dependence of stem cell maintenance in dauer, which is what we show.

NOTE: Figure provided for reviewer has been removed. It showed part of Figure 1 from Narbonne, P., Roy, R. (2006) Inhibition of germline proliferation during *C. elegans* dauer development requires PTEN, LKB1 and AMPK signalling. *Development* 133, 611-619. doi: <https://doi.org/10.1242/dev.02232>. We have removed unpublished data that had been provided for the referees in confidence.

Fig. 1 of Narbonne and Roy, 2006

(F) In both *daf-2(e1370)* and *daf-7(e1372)* larvae, germ cells arrest in mitotic interphase during dauer (arrowhead), as shown in a DAPI-stained, extruded *daf-7(e1372)* dauer gonad. (G) In *daf-2(e1370) glp-1(e2141)* dauer larvae, germ cell nuclei arrest in late stage meiotic prophase I, the nuclear size and chromosome morphology being consistent with the pachytene stage (inset)...

Narbonne and Roy also reference a quiescence of cell division with *constitutively active* Notch signaling, but this is also not a test of whether the germline depends on Notch signaling to stay in the undifferentiated, stem-like state:

“However, ongoing cellular divisions are never observed in the germ line of dauer larvae, even in *glp-1(oz112)gf* mutants (data not shown) in which Notch signalling is constitutively active (Berry et al., 1997).”

We see no evidence that Narbonne and Roy (2006) test whether dauer germ cells are maintained in a stem-like state while *glp-1(-)* in dauer itself, that is, whether they retain the ability to recover mitotic proliferation after return to the permissive temperature, not even a statement referencing “data not shown”. This is what we show in what is now Fig. 4.

Apart from that, the critical question would seem to be the nature of the Notch-independent quiescent state and that does not seem to be addressed.

We appreciate the Reviewer's question. In this revision, we clarify the connection between the germ cell cycle slowdown we observe upon pre-dauer starvation and the previously articulated protective mechanism acting in transiently starved adults on page 9:

Thus, the dauer germline is in a Notch-independent, G2-arrested state and can maintain germline stem cells even in the absence of niche signaling. Because germ cells cannot directly transition from mitotic G2 to prophase I of the meiotic cell cycle without completing mitotic M phase (Fox and Schedl 2015), cell-cycle quiescence protects dauer germ cells, like those in transiently starved adults (Seidel and Kimble 2015), from differentiating in the absence of Notch activity.

We find that onset of starvation causes the niche to become rapidly depleted of Notch ligand and germ cells slow their proliferation to arrive in G2 arrest.

(3) I found the reasoning of the later Results sections, 322-end, hard to follow. I am not persuaded by what I read. Downregulation of LAG-2/ligand in the Distal Tip Cells (DTCs), the germline niche, in L4 larvae response to adverse environmental conditions has already been established (Pekar et al., Development 2017) in some molecular detail at the level of transcription, so the reduction of LAG-2 in the DTCs in L2d is not in itself novel.

In our reply to Reviewer 1, we explain how we appropriately cite Pekar et al. 2017 and describe how our studies differ (and see Table). Pekar et al. looked at transcription driven by an incomplete *lag-2* promoter fragment, never at LAG-2 protein. That study showed that the DTC can respond to changes in the environment via an ASI/*daf-7*->DTC/*lag-2* axis in the L3-L4 stage during which development is continuous, not in animals in L2d, dauer, or recovering from dauer, the stages which we focus on. The time scale upon which Pekar et al. analyze *lag-2* transcriptional response to poor conditions is >12 hours (it is described only as endpoints in "late L4 animals reared from early L3"), while we analyze both increases and decreases in LAG-2::mNG in as little as 1 hour after environmental exposure changes. The mechanism of downregulation identified by Pekar et al. requires *daf-7*, while the mechanism of rapid LAG-2 protein depletion in the DTC that we discovered is *daf-7*-independent.

Most importantly, our evidence does not support a transcript-based regulation for the LAG-2 response we discovered (Fig. 5 and Results section pertaining to it, and see next reply), so we are not simply seeing another instance of the phenomenon observed by Pekar et al. We have gone on to add analyses of the *daf-7(e1372)* mutants which were found by Pekar et al. to have lower *lag-2* transcription in the DTC at baseline and to lack downregulation of *lag-2* transcriptional reporters when animals were exposed to poor conditions. We made several findings relevant to this question from the Reviewer, which we enumerate here with reference to where they appear in our manuscript.

1. First, baseline (fed L2) expression of *lag-2* transcriptional reporter and LAG-2::mNG protein are both lower in *daf-7(e1372)* mutants than in the otherwise wild-type control and in *daf-2(e1370)*. Thus we show that our reagents recapitulate the lower baseline *lag-2* expression in *daf-7(e1372)* mutants that Pekar et al. described for the later L4 larval stage. These results are shown in Fig. 3E-I.
2. Second, we find that *daf-7(e1372)* mutants show the same LAG-2 protein dynamics upon removal from food in L2d as *daf-2(e1370)*: rapid (within 1h) and substantial (~25-50%) decrease in LAG-2 protein signal in the DTC. These results are shown in Figure 3K-M. The natural conclusion from this result is that *daf-7* (like *daf-2*) is not required for the drop in LAG-2 protein signal upon food removal, and this phenomenon is therefore distinct from the *daf-7* dependent one reported by Pekar.
3. Finally, we observe the surprising result that, in the dauer stage, *daf-7(e1372)* mutants have dramatically *more* *lag-2* transcriptional activity than otherwise wild type worms. These results are shown in Fig. S3B. This result is surprising, because it implicates *daf-7* in the *negative* transcriptional regulation of *lag-2* (at least via its 3kb upstream promoter) in the DTC during dauer, a developmental stage at which *daf-7* signaling is thought to be low.

What appears to be novel is the argument that the regulation is occurring post-translationally. However, the experiments giving rise to this conclusion have caveats not addressed. While it is

commendable that the authors addressed caveats that might have been serious if they had only relied on a highly artificial transcriptional *lag-2* multicopy transgene reporter by looking at a polycistronic T2A knock-in expressing a co-regulated fluorescent histone, it was unclear to me how perdurance of the histone was accounted for; this would seem to be an important caveat of this approach.

We thank the Reviewer for this thoughtful comment. We have added an analysis of this reporter's signal in the somatic gonad lineage earlier in development when Z1.a and Z4.p divide to give rise to a *lag-2*-expressing DTC and a sister cell that does not express *lag-2*. We observe rapid signal decay (half life of ~3 hours) in the cell that does not express *lag-2*, and an average 4x greater signal in the DTC relative to the sister cell even at the earliest time point we capture after their division; within 3 more hours, the DTC has an average of ~200x more signal than its sister cell). This fluorescent protein turns over relatively quickly and signal increase from new protein production is easily detected.

This concurs with what was observed for this reagent in its initial publication (Medwig-Kinney et al., 2022), in which some specimens undergoing lateral inhibition in the AC/VU fate decision were observed to have equal levels of H2B::mTurquoise2 and others had notably different signal intensities in that same time window. In both the original study and our new experiment, the H2B::mTurquoise2 signal detectably diverges simultaneously with cells taking on *lag-2*(+) or *lag-2*(-) fates (to the experimentally relevant level of precision).

The perdurance of the fluorescent reporter is a concern primarily for observing downregulation. Long-lived fluorescent proteins could indeed mask a sudden drop in new protein production, and this is why we do not comment on the transcriptional reporter activity after food removal in Fig. 3 (see Fig. S3C). It is less clear how fluorescent protein perdurance would mask a rise in new protein production, which is what we are measuring with the polycistronic histone reporter in Fig. 5. The only way for the perdurance of the pre-existing fluorescent histone to mask the magnitude of increase in new protein production that we measure for LAG-2::mNG would be for the baseline to be so high that no *proportional* increase can be detected despite a large absolute increase. We therefore calculated the *absolute* increase as well, and show in Fig. S4 that no absolute increase is measured in any of the transcriptional reporters out to 8 hours of dauer recovery.

I will note that Pekar et al., 2017, which is cited by the Reviewers as a landmark study in this field, used membrane-localized fluorescent proteins made by transgenes driven by 3kb *lag-2* partial promoters (similar to the two membrane-fluorescence transcriptional reporters that we use), so the standard of evidence we provide is certainly no less than that highly respected study provided.

I am unconvinced that the level of regulation is at the level of protein trafficking or stability, which would seem to require expressing fluorescent LAG-2 from heterologous sequences. The use of endogenously tagged proteins generated by CRISPR/Cas to study protein trafficking and stability is widespread; such experiments do not require expressing the protein from heterologous sequences generally, and it is not clear why it would be required in this case. We are not arguing that transcriptional regulation is unimportant for *lag-2* in the DTC; we are demonstrating that transcription alone cannot account for the rapid dynamics of the LAG-2::mNG protein.

(4) I also did not understand the contention that somatic and germline development are "decoupled" if the mechanism includes modulation of the stemness cue and an independent imposition of both somatic and germline quiescence. Perhaps I'm not understanding what evidence there is that the imposition of quiescence is different for the somatic gonad and germline--they state that Cassada and Russell (1975) showed that the germline division stopped before somatic development but to me, that is not really decoupling, it is that they both stop at the tissue-specific appropriate checkpoint.

We thank the Reviewer for this astute comment. We apologize for being unclear. We intended to reference Cassada and Russell 1975 for the original observation that dauer animals slow but do not arrest somatic development in response to starvation. We discover here that L2d worms cease germline proliferation under starvation. We have de-emphasized the argument about decoupling since it became a sticking point. We have discovered that, while the soma will continue to its dauer-entry "checkpoint" regardless of nutritional state, the germline can stop cell divisions nearly immediately at any point upon the withdrawal of food. This is what we mean when we say the

response to starvation in L2d of these two tissues is decoupled, but we can explore this phenomenon without using that language.

Reviewer 3

SUMMARY OF THE ADVANCE MADE IN THIS PAPER AND ITS POTENTIAL SIGNIFICANCE TO THE FIELD

The manuscript submitted by Koitz et al. comprises a string of interesting observations that link various starvation regimens to the growth and proliferation of the gonad, and more particularly, to the germ line. The authors claim these regimens result in an uncoupling between the somatic gonad and germ line growth/division program, while also delineating several distinct means of interrogating the various signals and their effects on the intrinsic growth programs that drive the development of the germ line during the dauer diapause stage. In a second part of the manuscript the authors then focus their attention to how nutritional status and these various feeding/starvation regimens affect the levels of the Notch ligand LAG-2 in the dauer larva. Their final experiments drive the authors to conclude that the germ cells are held in quiescence in a Notch independent manner during the dauer stage. Although the observations are well described and quantified, I find that some of the authors' conclusions are not particularly well founded and would require significantly more experiments to corroborate their claims.

We appreciate that the Reviewer recognizes that our results are well described and quantified.

SUGGESTIONS TO AUTHORS

The work that is submitted is indeed interesting, but the rationale, the presentation, and the description/discussion of their various feeding regimens is quite confusing at times, while it often lacks any discussion of biological or physiological relevance. Perhaps more disappointing is the lack of any attempt to better understand these phenomena with the available genetic or cell/developmental biological tools. As a result, the manuscript reads as a list of interesting observations that are largely descriptive, without really providing any further insight as to how these phenomena are regulated or what their biological significance might be.

We regret that the Reviewer found our genetic and cell/developmental biological experiments confusing, and have taken greater care to explain how our findings lead to better understanding of the protection of germline stem cells during starvation and recovery. We also make sure to explain the profound biological and physiological relevance of maintaining germline stem cells and organismal resilience to starvation. As we say in lines 56-57: *"It is the preservation of fertility rather than increased total lifespan alone that makes dauer adaptive."* Surviving unfavorable conditions without the ability to reproduce is the same, evolutionarily speaking, as not surviving at all.

The designs of the experiments are generally straightforward (albeit sometimes difficult to follow). Most of the observations are quantitatively sound and the data are acquired with care and attention. As such, I find them mostly convincing, just complicated and confusing without reason. I have listed a number of points that I believe are important for the authors to address in order to reinforce their conclusions.

In our revised manuscript we have clarified the points of confusion the Reviewer shares below.

Lines 237-240 they describe an uncoupling of the soma and the germ line in these various feeding regimens. I think that there are other examples where the development of the soma and the germ line are uncoupled during various stresses, including dauer, but since the authors are convinced of this uncoupling in this context, it would be advantageous to show the evidence that the somatic cells are not affected by these feeding regimens without simply referencing Cassada. That might include comparing cell cycle timing etc...to demonstrate that the soma is not equally slowed and actually follows the germ cells.

This is an excellent point, which we address in Fig. 2E-F (formerly 2D-E) and its accompanying text on page 6, where we discuss the rate of germ cell proliferation:

Comparing the rate of germline growth for *daf-2(e1370)* animals revealed an even more dramatic response to starvation than that of germ cell number alone (Fig. 2E). Animals gained an average of one germ cell per hour while feeding (12h fed+24h fed), while animals that starved (12h fed+24h starved) added on average only one germ cell total in that 24 hour period (Fig. 2E, gray shaded box) and entered dauer more slowly (Fig. 2F, gray shaded box).

Because the dauers that are removed from food early enter dauer more slowly (that is, they spend more time in L2d), they are actually even more dramatically slowed in their germline proliferation than the absolute cell numbers reflect. They make fewer germ cells over even more time, while the soma makes the same number of cells, with different intercellular spacing depending on pre-dauer feeding (this is now discussed on page 4 and in Fig. S1).

Different dauer induction protocols changed overall gonad size but did not alter somatic cell numbers in the dauer gonad, only their proximity to one another (Fig. S1A-H), demonstrating that dauer gonads can vary in germ cell number and somatic cell size. The gonad displayed allometric scaling with body width and was relatively smaller in starved dauers compared to first-formed dauers (Fig. S1E), further demonstrating different growth dynamics in the soma and germline.

We do not simply cite Cassada for evidence of somatic progression.

The L1 diapause and the dauer diapause are not identical and the genes that are affected are not the same, although there may be some overlap. Although the germ cells present in the dauer larva do resemble L1 PGCs in their compacted state, this has not been described in dauer in any significant detail. The authors could use H3K9me3 or HPL-1 staining to evaluate if there are differences between the various regimens and between dauer larvae.

We understand that L1 and dauer are different, and do not imply otherwise. We know that germ cell chromatin compaction had not previously been described in dauer. We don't have the word count to fully develop this argument relating to L1, and so have removed it from the draft. We now simply cite prior work that reports dauer germ cells are known to be G2 arrested (Narbonne and Roy, 2006).

Lines 257-263 This is an interesting observation, but is only superficially supported by the work shown here. I think that to make this conclusion, much more work must be completed to compare somatic growth development and its scaling to the constraints imposed by the gonad or the germ cell population.

This comment references the following from the original draft:

256 We next asked if germ cell number was the only factor contributing to gonad size
257 differences. After progressive starvation, early wild-type (N2) dauers (Fig. 3E) displayed
258 greater gonad length (~1.7x) (Fig. 3F) and body width (~1.3x) (Fig. 3G) than late
259 dauers. However, the ratio of gonad length to body width also differs, with the early
260 dauer gonads being proportionally larger compared to the body width (~4.5:1) than
261 those of late dauer gonads (~3.5:1) (Fig. 3H), violating an assumption of simple
262 isometric scaling between body and gonad size. This suggests that growth is regulated
263 differently in the gonad vs. the rest of the body.

With these experiments, we demonstrate that gonad length does not scale isometrically with body width between what we now call “first-formed” and “starved plate” dauers with the measurements we show now in the supplement, Fig. S1A-D (above). Just because the measure is simple and elegant does not make it superficial. Without articulating suggestions, it is not clear how we could better demonstrate this easily measurable scaling difference.

Of course, more work could be done to explore the genetic and nutritional control of cell and organ scaling (a major, open question in developmental biology), but such work is not necessary to observe the difference in scaling. We also draw attention to our rather modest conclusion from this finding: “*The gonad displayed allometric scaling with body width and was relatively smaller in starved dauers compared to first-formed dauers (Fig. S1E), further demonstrating different growth dynamics in the soma and germline.*”

Line 292- "the" LAG-2 ligand

Germ line is a noun; germline is an adjective. Wild type is a noun; wild-type is an adjective...throughout

Thank you for this helpful comment. All usage will be corrected according to the *Development* style guide.

Lines 300-303 The text indicates that L1 PGCs are arrested in a notch independent manner is not presented, while no citation is provided.

We discovered that primordial germ cells of L1-arrested animals are Notch independent in this paper, formerly in lines 300-303. We removed these data from the study as they were deemed to be tangential.

Lines 320-321 This is a bit contentious. There is no evidence that this is fully Notch independent.

This and the following comments reflect the Reviewer’s confusion about tests of germline Notch-dependence. I will explain here and reference in subsequent replies.

The first experiment leverages a well-established reagent in the field, the *glp-1(bn18)* temperature-sensitive loss-of-function allele of the GLP-1/Notch receptor. This receptor is required for germ cells to remain in the mitotic, undifferentiated state during normal development (Kodoyianni et al., 1992). At the permissive (low) temperature, Notch is functional, but at the restrictive (high) temperature, Notch is not functional. The original manuscript described this in lines 292-321, and we have conducted a thorough revision to ensure we communicate the experimental paradigm clearly on page 8:

We investigated this question using worms carrying a temperature sensitive allele of *glp-1/Notch*, *glp-1(bn18)*. At the permissive temperature (16°C), *glp-1(ts)* mutants have adequate Notch signaling to remain fertile, but upon shifting to the restrictive temperature (25°C), they lose active Notch signaling and thus lose the germline stem cell population to differentiation (Kodoyianni et al. 1992). In adults, signs of meiotic entry appear in distal germ cells within 6 hours of temperature upshift (Fox and Schedl 2015), so the germ cell fate response to lost Notch-signaling is rapid.

To help contextualize these experiments, we give the following three examples of how temperature sensitive *glp-1* alleles have been used to abrogate Notch activity for many years by many workers.

From Kodoyianni et al., 1992:

The 15 recessive *glp-1(lf)* alleles that we have examined can be grouped into three phenotypic classes...

2) Six *glp-1* alleles (*q224*, *q231*, *e2141*, *e2144*, *bn18*, and *sy56*) are temperature sensitive (*ts*) in both germline and embryo (Table 1). When raised at permissive temperature (15°C), *glp-1(ts)* mutants appear wild-type, but when newly hatched larvae are shifted to restrictive temperature (25°C), germline induction fails and hermaphrodites are sterile. The germline effect of the *glp-1(ts)* mutations mimics that of putative *glp-1* null alleles.

Fox and Schedl 2015 report:

“To experimentally manipulate GLP-1 activity, we raised *glp-1(bn18)* mutants at permissive temperature (15°) and shifted synchronized adult animals to restrictive temperature (25°). ... Whereas there was no appreciable change in wild-type controls, in *glp-1(bn18)* mutants all PZ cells lost proliferative fate and initiated meiotic prophase within 10 hr at restrictive temperature (Figure 3A).”

A similar *glp-1(ts)* allele was used by Seidel and Kimble, 2015:

We used the temperature-sensitive *glp-1* allele *q224ts* to test whether GLP-1/Notch signaling is similarly required for maintenance of germline stem cells under starved conditions. The *q224ts* allele is the strongest of all known temperature-sensitive *glp-1* alleles and behaves like a null at the restrictive temperature (Austin and Kimble, 1987; Kodoyianni et al., 1992).

It is therefore not at all contentious to claim that our experiment rearing young larval *glp-1(bn18)* animals at the restrictive temperature for 24 hours causes loss of GLP-1 function, and biological processes that occur in that context can be said to be independent of that pathway's function.

We use the *glp-1(bn18)* temperature-sensitive allele to test whether the germ line in dauer animals depends on active Notch signaling to maintain an undifferentiated cell fate. We put these *glp-1(bn18)* worms into dauer by starving them at the permissive temperature, shifted them as dauers to the restrictive temperature for 24 hours, SDS treated the population to ensure only dauers were recovered, and put these dauer animals on food to complete development at the permissive temperature. After recovery to adulthood, 29/29 of these animals were fertile, while 27/28 well-fed control larvae were sterile (one had a few offspring, see Fig. 4A-B). We conclude, in accordance with ample precedent in the field, that the non-dauer germ line is dependent on Notch to maintain an undifferentiated, stem-like population (as expected). On the other hand, the dauer is able to maintain germ cells in the stem-like state despite the nonfunctionalization of the Notch-receptor. Thus, the dauer germ line is maintained in a Notch-independent state.

That the germ cells do not go into meiosis or exhibit meiotic features such as HIM-3 expression (was not shown) it would indicate that Notch is still blocking cells from engaging the full meiotic program and therefore must be functional. Perhaps there is another means of assessing Notch function at the molecular level to indicate that indeed it is extinguished during this state.

We are not testing whether Notch is active in this experiment, we are using the mutant background to ensure Notch is *not* active. As described above, Notch is nonfunctional as long as the *glp-1(bn18)* strain is maintained at 25°C (Kodoyianni et al., 1992), as evidenced by the sterilization of the non-dauer controls subjected to this treatment (Fig. 4). We do not need to stain for HIM-3 or other meiotic entry features, because we functionally test for the key feature of the successful prevention of premature meiotic entry: the presence of a fertile adult germ line.

The use of *glp-1(ts)* mutants to extinguish Notch signaling at the restrictive temperature is a long-established field standard (Kodoyianni et al., 1992, Seidel and Kimble 2015, Fox and Schedl 2015).

There is no reasonable basis for skepticism that it works to abrogate Notch receptor function in our experiments.

Lines 330-332 This observation suggests that the differences in the number of germ cells that are present are not a direct readout of LAG-2/GLP-1 signalling, but rather that LAG-2 is probably necessary, but is not the main constraint that dictates whether the cells proliferate or not.

We appreciate the opportunity to clarify. By the time the experiments discussed in this part of the text are performed, all animals are in the dauer stage and therefore are past the window in which germ cell proliferation was possible.

In our revised manuscript, we focus Fig. 3 on the rapid depletion of LAG-2::mNG after food removal in both *daf-2(e1370)* and *daf-7(e1372)*, showing that in the pre-dauer sensitive window when feeding affects germline growth, feeding also is required to maintain high levels of LAG-2::mNG in the DTC. Thus we implicate pre-dauer LAG-2 levels in pre-dauer germline proliferation, and note that LAG-2 protein in the DTC is kept low in dauer itself.

Line 354- insert that "the" dauer germ line...

Thank you for catching this typo!

Lines 441-445 Somatic gonad development clearly responds to dauer and/or starvation cues as it is much more compact. Individual cell size was not measured but there is clearly an adaptive scaling that takes place. So although the cell cycle uncoupling may take place as indicated by the authors. The germ line is under exquisite regulatory control, perhaps mostly by AMPK and PTEN, which largely adjusts growth and division to reflect the challenge, while presumably doing little to nothing to alter growth or division in the somatic gonad or the soma in general. This somatic regulation would stem from other cues that may ultimately impinge on one of the CDK inhibitors, which have significant effects on the somatic tissues.

We have refocused the paper on the DTC and germline, with less discussion of somatic growth generally. In short, most of what was in Fig. 3 has been moved to the Supplement or removed from the study. Since the decoupling framing was a sticking point for the Reviewers, we have de-emphasized it.

The authors use their imaging data to suggest that LAG-2 is under post-translational regulation to reduce its activity/abundance during the dauer stage. This should be elaborated in much greater detail, ideally by identifying the regulator and/or the stage-specific modification on the polypeptide itself by mass spectrometry.

Since we are observing a DTC-specific phenomenon, the proposed experiment would unfortunately not be informative. Even if we could purify LAG-2 protein from *C. elegans* dauers and perform mass spectrometry on that sample, most of the LAG-2 protein recovered would be from other cells (like neurons, Ouelett et al., 2008), not the DTC, as there are only two DTCs per animal, and our motivating finding is that LAG-2 protein expression is notable *low* in these cells in dauer.

Rest assured that elucidating the novel protein-based DSL-ligand regulatory mechanism-for which we here provide the first evidence-is an experimental goal of ours going forward, but this suggestion is well beyond the scope of the study.

Line 906 add a with after compared

Thank you for catching this typo!

It is unclear why the progressive starvation experiment was performed with a mixed population, whereas synchronized populations were used in the other described experiments. Would a synchronized population not make more sense for the starvation experiment, as you could ensure all animals were starved for the same duration? Perhaps the authors could clarify why they used a different population.

In the original manuscript, worms were synchronized only when a staged population was required for food removal experiments in strains forming constitutive dauers. Now in the revision, worms are synchronized in the first step of the liquid culture experiment (Fig. 1A-B, after Hibsichman et al., 2021). We have clarified each protocol in the Methods section. It is worth noting that for all of the experiments on plates (starvation, first-formed, high density), worms form dauers after >1

generation, so starting with a synchronized population would still yield a mixed population at the time of food exhaustion and dauer formation.

I am surprised the authors did not repeat at least part of the experiments in Fig 2 with the *daf-7* mutant. The data would be even more compelling to show that the potential decoupling they have revealed occurs regardless of the modality of dauer induction.

We thank the Reviewer for this excellent suggestion. We have augmented the *daf-7(e1372)* data that was originally in Fig. 1 with the germ cell marker and analysis of additional dauer formation protocols. We now include a complete food removal experiment on the *daf-7(e1372)* strain in Fig. 2 and Fig. S2. The *daf-7(e1372)* mutants show the same behavior that we observed for *daf-2(e1370)*: food-dependent germline growth. This demonstrates that *daf-7* is dispensable for nutritionally-sensitive germline proliferation in L2d. We have also added *daf-7(e1372)* to our *lag-2* expression studies in Fig. 3 and observe the same behavior as *daf-2(e1370)*: rapid depletion of LAG-2::mNG upon food removal. Additional observations about *daf-7*-dependent *lag-2* regulation now appear in Fig. S3. All of these experiments support our conclusion that decoupling occurs regardless of the modality of dauer induction, and in the absence of both wildtype *daf-2* and *daf-7*.

Fig 5A-B: I wonder if the authors could employ a qPCR method to directly measure levels of LAG-2 mRNA in a more quantitative manner. It would be more accurate than simply looking at a transcriptional reporter and would nicely complement their other data.

Unfortunately qPCR will not detect *lag-2* transcript from only the DTCs (which we demonstrate is quite low), since *lag-2* is also expressed neuronally in dauer (Ouelett et al. 2008).

The image resolution of some of the figures seem low. In Fig 1 for example, some of the graph elements like the axes and points are much blurrier than the labels.

Thank you for pointing this out; we will make sure that final figure uploads have appropriate resolution.

Second decision letter

MS ID#: dev.204972R1

MS TITLE: *C. elegans* pre-dauer starvation arrests germ cell proliferation and ligand presentation by the germline stem cell niche via a rapidly reversible, protein-based mechanism

AUTHORS: Fred A. Koitz, Camille P. Miller, Brian Kinney and Kacy Lynn Gordon

Dear Dr Gordon,

I have now received all the referees reports on the above manuscript, and have reached a decision. The referees' comments are appended below.

Overall, reviewers found that your revision addressed many of their comments, but they also still find there to be outstanding areas of improvement. Particularly, reviewers are concerned that the focus on post-translational regulation is not fully supported. As Reviewer 1 points out, post-transcriptional regulatory mechanisms have not been ruled out, and a fuller consideration of all possibilities is needed. In view of this, the title of the paper should be reassessed as well. Reviewers also suggest reanalysis of images and statistical tests to strengthen claims. Altogether, the comments don't require you to perform new experiments but to ensure that the claims made are robustly supported by the data provided.

We invite you to submit a revised manuscript in Development, provided that the referees' comments can be satisfactorily addressed. Please attend to all of the reviewers' comments in your revised manuscript and detail them in your point-by-point response. If you do not agree with any of

their criticisms or suggestions explain clearly why this is so. If it would be helpful, you are welcome to contact us to discuss your revision in greater detail. Please send us a point-by-point response indicating your plans for addressing the referees' comments, and we will look over this and provide further guidance.

Reviewer 1

Advance summary and potential significance to field

In this revised manuscript by Koitz et al., the authors provide evidence to show that pre-dauer feeding determines germ cell number, which correlates with brood size in post-dauer adults. They also provide experiments to show the regulation of LAG-2 in the DTC during dauer and the requirements of Notch during germline quiescence and recovery. The revisions have greatly improved the clarity of the text and the figures. I do have some additional comments on the revised manuscripts.

Comments for the author

1. I will concede the point that lag-2 expression has not been examined specifically in the distal tip cells of dauers. I would highly recommend that the authors discuss what related experiments have been done previously in the manuscript, because I am sure that other scientists in the field also think that this experiment has been done. Related to this point, the authors conclude that post-translational regulation of LAG-2 is occurring during dauer. I am convinced that lag-2 transcription is occurring consistently through dauer, but I am not convinced that post-translational regulation is the mechanism regulating LAG-2 protein levels. Regulation at the translational level, through siRNAs or miRNAs, could also account for the altered LAG-2 levels. Without further experimentation, I do not think the authors could distinguish between the two explanations.
2. Figure 1 B, C: In wild type, the number of germ cells in dauer is not strictly correlating with brood size in adults. First-formed dauers have significantly more germ cells than high-density dauers, but they appear to have equal numbers of progeny in adulthood. We also see that the significantly decreased brood size in post-dauers from starved plates compared to never-dauer as reported by several groups is not repeated here. What would be the explanation for these observations? The authors should comment on how these results do not support one of their main conclusions.
3. Figure 1 F, G: The authors state that DAF-2 and DAF-7 pathways are not required for the food-dependent regulation of germ cell number in pre-dauers. However, this conclusion is hard to judge in F and G. The authors should provide statistical analysis of figure 1 F and G similar to what is shown for B and C. Additionally, in wildtype dauers, do high-density dauers have significantly greater germ cell numbers than starved plate dauers? If first-formed and high-density dauers do not have differences in germ cell numbers in daf-2 and daf-7 mutants as suggested by their comparisons to constitutive dauers, that result would suggest that DAF-2 and DAF-7 do play a role in pre-dauer germ cell number.
4. Figure 3 and 5: the images presented in figures 3 and 5 are obviously different in transgene expression and support the conclusions in the manuscript. The methods state that all images were taken with the same settings. I have a concern that some of the images are not in the linear range, and therefore, cannot be quantified accurately. I do not intend for any experiments to be repeated, just that care should be taken when comparing fold differences in expression across strains.
5. Figure 4: the experiment with temperature-sensitive notch during dauer recovery is very interesting!

Reviewer 2

Advance summary and potential significance to field

All three reviewers were highly critical of the original submission, with many overlapping concerns. As a result of the authors' efforts to address these concerns, the paper has been extensively revised

and the major points that are left were clarified, and it was much easier to read and understand what the authors did and how they interpreted their data.

With the findings more clearly presented, rather than adding responses to individual points in the 18-page rebuttal, I read the paper afresh to see if it is "a significant and novel contribution to our understanding of developmental mechanisms", as that is the ultimate test of whether a manuscript is suitable for Development. Even if I put aside my continued reservations about whether the post-translational regulation of LAG-2 has been adequately established by the approach taken here without further manipulation and testing--the main finding of the paper--that finding would itself be incremental, as there is no mechanistic insight into how such regulation may occur or even whether it is causal. I simply am not persuaded that this paper is appropriate for Development in its current form and judge it as more appropriate for a more archival journal.

Reviewer 3

Advance summary and potential significance to field

The authors have addressed most if not all my concerns with the original submission, the most critical of which was the unsubstantiated claim that initially reported observations were consistent with a decoupling of somatic and germline development/growth. The other issues were argued in a manner that satisfied any other concerns I might have had with their conclusions.

The manuscript is now distilled into a collection of observations that show convincingly that the life history, namely the resources available to the larva before entering dauer, dictate the number of germ cells that will be present in the dauer stage. This is independent of the manner (signalling pathway) that is disabled to trigger dauer formation. Secondly, the germ cells are maintained in a quiescent, non-meiosis committed state during the dauer stage in a manner that is independent of Notch signalling. This is reminiscent of work published by Seidel and Kimble who indicated that some other means of maintaining quiescence must be active in animals undergoing quiescence. Lastly, the signal of LAG-2/Delta is reduced as the animals enter the diapause and this is not due to changes at the transcriptional level. Furthermore, as animals recover, the protein levels increase but transcriptional levels remain constant. The lack of LAG-2 during the dauer stage probably does not account for the quiescence, but could provide the Notch dependence to the germ cells as the animals re-activate Notch-dependent regulation of the germ cell divisions.

These three points are all additions to our current knowledge of how Notch contributes to germ/stem cell homeostasis and how it is regulated during periods of environmental challenge.

Comments for the author

Minor issues:

Line 65-Corsi reference is repeated

Line 87-some refs are italicized while most are not

Lines 323 and 330-germline is an adjective; germ line is a noun

Line 372- Maybe i am missing something with this construct and as much as I appreciate the lengths the authors have gone to control for transcription, I am a bit at odds as to why they did not use a tagged LAG-2 here so that the protein levels could have been assessed directly alongside the transcriptional baseline. It seems like a missed opportunity to address most of the concerns about transcription vs protein based regulation in one experiment. Nevertheless, in combination with the endogenous NG labelled LAG-2 experiment, I am convinced that this is not a transcriptional response and depends on changes incurred on the LAG-2 protein.

The Discussion is succinct and that is good; it matches the manuscript body, but the focus on p53 or cyclin D seems to suggest that these critical growth regulators follow some pattern. The truth is that there are likely thousands of proteins that undergo translational regulation that are not oncogenes or involved in growth per se; I think focusing on cyclin D and p53, without indicating that several proteins use this same regulatory modality is a bit misleading, or at least, not really objective.

Second revision

Author response to reviewers' comments

ABRIDGED EDITORIAL DECISION OF 12/2/2025

Dear Dr Gordon,

I have now received all the referees reports on the above manuscript, and have reached a decision. The referees' comments are appended below.

Overall, reviewers found that your revision addressed many of their comments, but they also still find there to be outstanding areas of improvement. Particularly, reviewers are concerned that the focus on post-translational regulation is not fully supported. As Reviewer 1 points out, post-transcriptional regulatory mechanisms have not been ruled out, and a fuller consideration of all possibilities is needed. In view of this, the title of the paper should be reassessed as well. Reviewers also suggest reanalysis of images and statistical tests to strengthen claims. Altogether, the comments don't require you to perform new experiments but to ensure that the claims made are robustly supported by the data provided.

[...]

I look forward to receiving your revised manuscript.

With best wishes,
Mansi Srivastava
Handling Editor

Dear Dr. Srivastava,

Thank you for your decision, and for sharing the Reviewers' additional comments. We have addressed these with attentive editing of the manuscript, two new additional supplemental figures, a new supplemental table, and a more thorough explanation of the rationale behind our choice of statistical methods. We have also changed the title in accordance with your suggestion.

Here we succinctly highlight the changes we made to address the concerns highlighted in the Editor's letter.

1. ...the focus on post-translational regulation is not fully supported. As Reviewer 1 points out, post-transcriptional regulatory mechanisms have not been ruled out, and a fuller consideration of all possibilities is needed.

Please see our illustrated response to Reviewer 1 point 1, and the accompanying new Figure S6. We believe this critique is based on a misunderstanding of the polycistronic reporter, which we have endeavored to explain more clearly in this response and in our revised manuscript. Our results are not consistent with regulation of LAG-2 protein abundance at the transcript level.

1b. In view of this, the title of the paper should be reassessed as well.

We agree it is best to change the title to: Pre-dauer starvation rapidly and reversibly reduces niche proliferative signaling to the *C. elegans* germ line

While we stand by our interpretation of our results, we think it is more specific to our findings to state in the abstract that the LAG-2 protein change is "independent of transcriptional upregulation" instead of "regulated at the protein and not transcript level"

2. Reviewers also suggest reanalysis of images

Please see our response to Reviewer 1 point 4, illustrated by new Figure S5. The Reviewer's concern about pixel saturation pertains to the display image only, not the data that were measured and analyzed.

3. Reviewers also suggest reanalysis of [...] statistical tests

We have included a new Supplemental Table S1 to report pairwise comparisons that don't fit on our graphs; sharing all of these was a helpful suggestion.

Please see our response to Reviewer 1 point 2 and an addition to the Methods section on statistical analysis. We present our rationale for using non-parametric tests because our data are not normally distributed and have unequal variances. Others in the field have used parametric tests (like Student's t-test), which are more sensitive, for their brood size comparisons. Using Student's t-test, we can recapitulate the findings others have made previously (the difference in brood size between the never-dauer controls and the starved-plate dauers). Importantly, we never make a claim about this brood size comparison. We present our findings and the results of our statistical analysis honestly: the brood size differs significantly between animals that were severely starved before dauer and those that were not severely starved before dauer.

Thank you for the opportunity to revise. We hope this version of the manuscript is acceptable and are happy to discuss and make further changes if it is not.

Point-by-point begins below.

Thank you,

Kacy Gordon, on behalf of my coauthors

Comments from the Reviewers:

Reviewer 1: SUMMARY OF THE ADVANCE MADE IN THIS PAPER AND ITS POTENTIAL SIGNIFICANCE TO THE FIELD

In this revised manuscript by Koitz et al., the authors provide evidence to show that pre-dauer feeding determines germ cell number, which correlates with brood size in post-dauer adults. They also provide experiments to show the regulation of LAG-2 in the DTC during dauer and the requirements of Notch during germline quiescence and recovery. The revisions have greatly improved the clarity of the text and the figures. I do have some additional comments on the revised manuscripts.

We thank the Reviewer for their suggestions, and appreciate the feedback that helped us improve the clarity of our manuscript.

SUGGESTIONS TO AUTHORS

1. I will concede the point that lag-2 expression has not been examined specifically in the distal tip cells of dauers. I would highly recommend that the authors discuss what related experiments have been done previously in the manuscript, because I am sure that other scientists in the field also think that this experiment has been done.

We thank the Reviewer for pointing out some important literature in their first Review, and we include the following references to highlight prior studies of *lag-2* expression in dauer and under other types of environmental stress. Especially see Discussion lines 481-500.

Pekar et al., 2017 (*lag-2* transcriptional reporter response to environment in L4 larvae, cited throughout)

Dalfó et al., 2012 (*lag-2* transcriptional reporter response to the environment in young adults, after treatment starting in L3, cited throughout)

Narbonne & Roy, 2006 (*lag-2* transcriptional activity, germ line quiescence for proliferation in Notch GOF, germ line quiescence for germ cell differentiation in Notch LOF during dauer, cited throughout)

Ouellet et al., 2008 (*lag-2* in the dauer nervous system, cited in Discussion)

Karp and Greenwald, 2013 (*lag-2* transcriptional reporter activity in the dauer VPCs, cited in Discussion and in Methods when we describe transcriptional reporters).

Related to this point, the authors conclude that post-translational regulation of LAG-2 is occurring during dauer. I am convinced that *lag-2* transcription is occurring consistently through dauer, but I am not convinced that post-translational regulation is the mechanism regulating LAG-2 protein levels. Regulation at the translational level, through siRNAs or miRNAs, could also account for the altered LAG-2 levels. Without further experimentation, I do not think the authors could distinguish between the two explanations.

The Editor highlights this particular concern of the Reviewer's. We suspect the sticking point comes from confusion caused by our failure to adequately explain how the polycistronic reporter is made and how it functions molecularly. This reagent is not our invention, it is a published reagent (one of many using the same mechanism), and we now explain it in more depth in lines 367-378 and below in some diagrams, and have verified its coregulation with *lag-2* in Supplemental Figure S6.

In addition to two traditional *lag-2* promoter-driven transgenes (transcriptional reporters) and an endogenously tagged LAG-2::mNG CRISPR allele (protein tag), we use another previously published reagent (Medwig-Kinney et al., 2022). This reagent is a polycistronic knock-in of a fluorescent histone H2B::mTurquoise2 gene at the endogenous *lag-2* locus: *lag-2(bmd202[lag-2::P2A::H2B::mTurquoise2^{lox5111}^2xHA])*. Downstream of the *lag-2* coding gene, the STOP codon is replaced by a CRISPR/Cas9-mediated knock-in of a P2A sequence, then a histone coding sequence fused to the coding sequence of mTurquoise2.

Endogenous *lag-2* genomic locus

From these sequences, a single mRNA is transcribed, encoding LAG-2, P2A, and the fluorescent histone.

This is the transcript species that encodes the visible reporter that we measure, H2B::mTurquoise2. All H2B::mTurq2-encoding mRNAs have the *lag-2* coding and UTR sequences present with them in a single molecule. Any transcript-based regulation—delayed translation, mRNA degradation, binding by RNA binding proteins, RNA interference by endogenous siRNA or microRNA or exogenous RNAi (see new experiment below)—that affects native *lag-2* will also affect the visible reporter, H2B::mTurquoise2. Distinct transcript-based regulation of these two gene products is not possible in this strain because there is just one transcript species encoding them both.

When this mRNA is translated, the viral-derived P2A sequence causes the ribosome to skip a peptide bond in the growing polypeptide chain in what is called ribosome skipping, StopGo translation, or Stop-Carry On translation. This mechanism was discovered in 2001 (Donnelly et al., 2001, *J Gen Virol.*). Such so-called “self-cleaving” peptides are found in many viruses and have been used in many transgenic constructs (reviewed by de Lima and Lanza, 2021, *Viruses*). In our reporter, it leads to the cotranslation of a visible H2B::mTurq2 marker in a 1:1 stoichiometric ratio with the untagged LAG-2 native protein product.

Thus, the fluorescent histone that we measure is coregulated at the transcript level with *lag-2* until translation is complete. At that point, the proteins are entirely distinct and can be differentially regulated (translocation to different compartments within the cell, degradation at different rates, different protein-protein interactions, etc.). We improved our description of this process in lines 367-378.

If LAG-2 abundance were being regulated at the transcript level, the H2B::mTurq2 would show the same abundance dynamics as LAG-2::mNG, which we do not observe (Fig. 5). The divergence between the tagged LAG-2::mNG and this unique polycistronic H2B::mTurquoise2 reporter drives our conclusion that the abundance of LAG-2::mNG that we measure is being regulated post-translationally. We believe that this conclusion is inescapable, and we cannot honestly propose a role for transcript-based regulation that could escape detection with this reagent.

We consider this P2A knock-in to be a validated method to produce a cotranslated reporter. Additionally, we have now experimentally demonstrated that this construct indeed is coregulated with *lag-2* at the transcript level. We have added a control experiment in which we use *lag-2* RNAi to knock down the visible H2B::mTurquoise2 reporter (Fig. S6). We have included it here for ease of reference. As predicted, 24h of larval feeding of *lag-2* RNAi (~900 bp of sequence of *lag-2* exon 1, intron 2, and part of exon 2) has no effect on the *lag-2p::mNeonGreen::PH* transcriptional reporter (Fig. S6E) but causes a ~44% knockdown of H2B::mTurquoise2 in the nucleus (see below in cyan and quantification in Fig. S6D).

A LAG-2 and H2B::mTurq2 encoded by a single polycistronic mRNA

If there are continued concerns that unrecognized transcript-based regulation of *lag-2* can be taking place, we ask that they identify specific flaws in our reasoning or understanding of our reagent.

We have taken the claim that the LAG-2 regulation is protein-based out of the title and abstract.

2. Figure 1 B, C: In wild type, the number of germ cells in dauer is not strictly correlating with brood size in adults. First-formed dauers have significantly more germ cells than high-density dauers, but they appear to have equal numbers of progeny in adulthood [...see next...] The authors should comment on how these results do not support one of their main conclusions.

The Reviewer makes an astute point. In unmated *C. elegans* hermaphrodites like these, fecundity is limited by self-sperm production, in which a subset of the L4 larval germ cells differentiate as sperm. Self-sperm production is compromised in recovered starvation dauers (Ow et al., 2021, Gimond et al. 2025). If first-formed and high-density dauers make different numbers of pre-dauer germ cells, but both make enough germ cells to execute normal spermatogenesis in the post-dauer L4 stage, they will not necessarily go on to have different numbers of offspring.

We have carefully inspected the manuscript to ensure we never claim that germ cell number in dauer is proportional to eventual post-dauer brood. That is not one of our main conclusions.

We also see that the significantly decreased brood size in post-dauers from starved plates compared to never-dauer as reported by several groups is not repeated here. What would be the explanation for these observations?

In Fig.1B, the brood size of the recovered starved-plate dauers is lower in its mean and different in its distribution from the never-dauer control, with a long tail of sub-normal fertility. Brood size means: never-dauer=244, starved plate=181, a difference of ~25%. This difference is comparable to the mean difference reported between never-dauer and recovered starvation dauers by Ow et al. (2021) of ~15%. Student's t-test finds $t=2.130$, $p<0.05$ for the comparison between never-dauer control and starved dauer brood alone. However, this difference is not robust to correction for multiple comparisons, and this isn't the appropriate test for non-normally distributed data. While we fail to find a statistically significant difference using our preferred method of analysis (see following), we certainly never claim these brood sizes are the same.

Our data are not normally distributed (with the aforementioned long tail), which violates the assumptions of an ANOVA. We therefore use a non-parametric Kruskal-Wallis test, and non-parametric tests have less power to detect statistically significant differences. Though ANOVA has been found to be robust to violations of the assumption of normality, it is not robust to unequal variances (Blanca et al., 2017). The starved plate dauer data has ~12x the variance of the never-dauer control data (F test to compare variances $F_{16, 14}=12.15$ $p<0.0001$). We report results from what we believe is the most appropriate statistical test for the data we have, and we justify these choices in the Methods in lines 893-902.

We make no claims about the difference in brood between the never-dauer controls and the post-starvation dauers. We report our robust findings that the broods of post-starvation dauers are indeed significantly different from the high-density dauers and from first-formed dauers (which results we have now included in Supplemental Table S1A).

3. Figure 1 F, G: The authors state that DAF-2 and DAF-7 pathways are not required for the food-dependent regulation of germ cell number in pre-dauers. However, this conclusion is hard to judge in F and G. The authors should provide statistical analysis of figure 1 F and G similar to what is shown for B and C.

We thank the Reviewer for the helpful suggestion and have now added a supplemental table (Table S1) showing all of the pairwise comparisons in Fig.1 B, C, F, and G. Putting each pairwise comparison on the graphs was not possible, but we appreciate the Reviewer's point that comparable results should be shared between the wildtype and mutant conditions.

Additionally, in wildtype dauers, do high-density dauers have significantly greater germ cell numbers than starved plate dauers?

We thank the Reviewer for catching that we left off a comparison between starved plate and high-density germ cell numbers in wildtype shown in Figure 1B. We have added this ($p<0.0001$, ****), and it also appears in Table S1. We have restructured the graph and corrected a labeling error (liquid and starved plate labels had been mistakenly swapped) to highlight the comparisons to the starved plate condition. Other comparisons appear in Table S1A.

If first-formed and high-density dauers do not have differences in germ cell numbers in *daf-2* and *daf-7* mutants as suggested by their comparisons to constitutive dauers, that result would suggest that DAF-2 and DAF-7 do play a role in pre-dauer germ cell number.

We have edited the manuscript to emphasize our argument that *daf-2* and *daf-7* are not required for germ cell number *to respond to nutrition* before dauer, not that they don't play a role in pre-dauer germ cell number at all. Indeed, Table S1C (bottom) makes clear that there is a difference in germ cell proliferation between the *daf-2(e1370)* and *daf-7(e1372)* genotypes under identical rearing conditions. Welch's t-test indicates mean germ cell numbers differ between constitutive

dauers of *daf-2(e1370)* and *daf-7(e1372)* genotypes. Welch's corrected $t=7.853$, $df=57.24$, Two-tailed $p<0.0001$. Such effects are to be expected for mutants in growth and development pathways. The key point of the experiments in Fig.1 is that there are "starved" pre-dauer conditions (liquid culture and starved plates) that generate relatively few germ cells-regardless of genotype-compared to the "fed" pre-dauer conditions of high-density plating, first-formed dauers, and (for the *daf-c* mutants) constitutive dauer formation in the presence of food. The title of this results section is clear in presenting this categorical rather than proportional difference: **Dauers that form in the presence of food have more germ cells and recover with larger broods than those that form after abject starvation.** We make no claims about similarities or differences between high-density vs. first-formed dauers. Our statistical analyses here are corrected for multiple comparisons, meaning our power is decreased by the number of treatment groups under investigation. Failing to identify a statistically significant difference between two groups is not the same as proving that the groups are identical.

4. Figure 3 and 5: the images presented in figures 3 and 5 are obviously different in transgene expression and support the conclusions in the manuscript. The methods state that all images were taken with the same settings. I have a concern that some of the images are not in the linear range, and therefore, cannot be quantified accurately. I do not intend for any experiments to be repeated, just that care should be taken when comparing fold differences in expression across strains.

The reviewer is correct that some pixels in the micrographs in Fig.3 and Fig.5 are shown saturated, but this is true only for the scaling in the displayed image; the raw images do not have saturated pixels. The 8-bit display dynamic range has 256 gray values, so to display the full range of LAG-2 intensities (across a ~40-fold range), we had to choose between showing saturated pixels at the high end or a featureless black field at the low end. Our raw 32-bit data dynamic range can include billions of gray values (far more than the human visual system can perceive). For example, the "8h recovery" DTC in Fig. 5C

contains a range of over 200,000 gray values just within the DTC ROI. Fluorescence intensity measurements were made on sum projections of raw images before the production of figure display panels (as described in our Methods).

We have added a supplemental figure Fig. S5 showing each image in Fig. 5C as displayed in that figure (Fig.S5A), with the same scaling but the LAG-2::mNG channel colored using the "HiLo" lookup table, which shows pixels with 0 value (blue) and saturated pixels (red) (Fig.S5B), and with that same "HiLo" lookup table but the original scaling of the 32-bit images (Fig.S5C), in which no saturated pixels are observed.

We would appreciate guidance from Development about the journal's standards for displaying images with dramatically different pixel values of the same fluorescent protein.

5. Figure 4: the experiment with temperature-sensitive notch during dauer recovery is very interesting!

We thank the Reviewer for their kind words! The advice of the Reviewers to restructure the paper certainly helped experiments like this one have more impact.

Reviewer 2: SUMMARY OF THE ADVANCE MADE IN THIS PAPER AND ITS POTENTIAL SIGNIFICANCE TO THE FIELD

All three reviewers were highly critical of the original submission, with many overlapping concerns. As a result of the authors' efforts to address these concerns, the paper has been extensively revised and the major points that are left were clarified, and it was much easier to read and understand what the authors did and how they interpreted their data.

With the findings more clearly presented, rather than adding responses to individual points in the 18-page rebuttal, I read the paper afresh to see if it is "a significant and novel contribution to our understanding of developmental mechanisms", as that is the ultimate test of whether a manuscript is suitable for Development. Even if I put aside my continued reservations about whether the post-translational regulation of LAG-2 has been adequately established by the approach taken here without further manipulation and testing--the main finding of the paper--that finding would itself be incremental, as there is no mechanistic insight into how such regulation may occur or even whether it is causal. I simply am not persuaded that this paper is appropriate for Development in its current form and judge it as more appropriate for a more archival journal.

We appreciate that the Reviewer read our revised manuscript and found it more clearly presented. While Reviewer 2 finds our results to represent incremental advances, Reviewer 3 (below) succinctly states, "These three points are all additions to our current knowledge of how Notch contributes to germ/stem cell homeostasis and how it is regulated during periods of environmental challenge."

SUGGESTIONS TO AUTHORS

None given

Reviewer 3: SUMMARY OF THE ADVANCE MADE IN THIS PAPER AND ITS POTENTIAL SIGNIFICANCE TO THE FIELD

The authors have addressed most if not all my concerns with the original submission, the most critical of which was the unsubstantiated claim that initially reported observations were consistent with a decoupling of somatic and germline development/growth. The other issues were argued in a manner that satisfied any other concerns I might have had with their conclusions.

The manuscript is now distilled into a collection of observations that show convincingly that the life history, namely the resources available to the larva before entering dauer, dictate the number of germ cells that will be present in the dauer stage. This is independent of the manner (signalling pathway) that is disabled to trigger dauer formation. Secondly, the germ cells are maintained in a quiescent, non-meiosis committed state during the dauer stage in a manner that is independent of Notch signalling. This is reminiscent of work published by Seidel and Kimble who indicated that some other means of maintaining quiescence must be active in animals undergoing quiescence. Lastly, the signal of LAG-2/Delta is reduced as the animals enter the diapause and this is not due to changes at the transcriptional level. Furthermore, as animals recover, the protein levels increase but transcriptional levels remain constant. The lack of LAG-2 during the dauer stage probably does not account for the quiescence, but could provide the Notch dependence to the germ cells as the animals re-activate Notch-dependent regulation of the germ cell divisions.

These three points are all additions to our current knowledge of how Notch contributes to germ/stem cell homeostasis and how it is regulated during periods of environmental challenge.

We thank the Reviewer for their helpful feedback and appreciate their summary of the impact of this work.

SUGGESTIONS TO AUTHORS**Minor issues:****Line 65-Corsi reference is repeated**

Fixed, thank you!

Line 87-some refs are italicized while most are not

Fixed, thank you!

Lines 323 and 330-germline is an adjective; germ line is a noun

Fixed, thank you!

Line 372- Maybe i am missing something with this construct and as much as I appreciate the lengths the authors have gone to control for transcription, I am a bit at odds as to why they did not use a tagged LAG-2 here so that the protein levels could have been assessed directly alongside the transcriptional baseline. It seems like a missed opportunity to address most of the concerns about transcription vs protein based regulation in one experiment. Nevertheless, in combination with the endogenous NG labelled LAG-2 experiment, I am convinced that this is not a transcriptional response and depends on changes incurred on the LAG-2 protein.

We agree with the Reviewer that it would be ideal to simultaneously image both the endogenously tagged LAG-2::mNG and the polycistronic H2B knock-in in the same cells, but we were not able to make the combination LAG-2-fluorescent protein fusion and polycistronic histone knock in at the endogenous *lag-2* locus. We are glad that the Reviewer is nonetheless convinced that we have robust and consistent results with both of these reagents individually, and the difference is so vast, with no transcriptional upregulation detected at all over 8 hours of recovery, that we can still conclude that the dynamics of the polycistronic reporter differ from those of the LAG-2::mNeonGreen.

The Discussion is succinct and that is good; it matches the manuscript body, but the focus on p53 or cyclin D seems a suggest that these critical growth regulators follow some pattern. The truth is that there are likely thousands of proteins that undergo translational regulation that are not oncogenes or involved in growth per se; I think focusing on cyclin D and p53, without indicating that several proteins use this same regulatory modality is a bit misleading, or at least, not really objective.

We appreciate this feedback on the new Discussion section. We have revised the section to remove reference to oncogenic mutation of p53 and cyclin D, and we end thus, in lines 429-440:

Such a regulatory paradigm famously governs the p53 family of tumor suppressor proteins, which are transcribed, translated, ubiquitinated, and degraded under normal growth conditions when p53 is not active. DNA damage or other stresses repress its ubiquitin ligase, allowing the rapid accumulation, nuclear translocation, and activity of p53 (Abueta et al., 2022). Recently, cyclin D has been discovered to be regulated by a similar process (Chaikovskiy et al., 2021; Maiani et al., 2021; Simoneschi et al., 2021). Other proteins that mediate rapid response to changing environmental conditions are also regulated by steady state production and degradation in the absence of induction, like the hypoxia inducible transcription factor subunit HIF- α (reviewed in Flick and Kaiser, 2012 and in Weidemann and Johnson, 2008) and the Nrf2 transcriptional regulator of heme oxygenase HO-1 (Stewart et al., 2003; reviewed in Flick and Kaiser, 2012).

Third decision letter

MS ID#: dev.204972R2

MS TITLE: Pre-dauer starvation rapidly and reversibly reduces niche proliferative signaling to the *C. elegans* germ line

AUTHORS: Fred A. Koitz, Camille P. Miller, Brian Kinney and Kacy Lynn Gordon

Dear Dr Gordon,

I am happy to tell you that your manuscript has been accepted for publication in *Development*, pending our standard publication integrity checks.